# Effects of decimetre-scale surface roughness on L-band Brightness Temperature of Sea Ice

Maciej Miernecki[2,1], Lars Kaleschke[3,1], Nina Maaß[1], Stefan Hendricks[3], and Sten Schmidl Søbjærg[4]

[1]Institute of Oceanography (IfM), University of Hamburg, Bundesstr. 53, 20146 Hamburg Germany

[2]Centre d'Etudes Spatiales de la Biosphère (CESBIO), 18 avenue Edouard Belin bpi 2801, 31401 Toulouse Cedex 9, France

[3]Alfred Wegener Institute, Helmholtz Centre for Polar and Marine Research, Bremerhaven, Bussestrasse 24, 27570 Bremerhaven, Germany

[4]Technical University of Denmark, Ørsteds Plads, 2800 Kgs. Lyngby Danmark

**Correspondence:** Maciej Miernecki (maciej.miernecki@cesbio.cnes.fr)

**Abstract.**

Sea ice thickness is an Essential Climate Variable. Current L-Band sea ice thickness retrieval methods do not account for sea ice surface roughness that is hypothesized to be not relevant to the process. This study attempts to validate this hypothesis that has not been tested yet. To test this hypothesis, we created a physical model of sea ice roughness based on geometrical optics and merged it into the L-band emissivity model of sea ice that is similar to the one used in the operational sea ice thickness retrieval algorithm. The facet description of sea ice surface used in geometrical optics is derived from 2-D surface elevation measurements. Subsequently the new model was tested with $T_B$ measurements performed during the SMOSice2014 field campaign. Our simulation results corroborate the hypothesis that sea ice surface roughness has marginal impact on near-nadir $T_B$ (used in the current operational retrieval). We demonstrate that the probability distribution function of surface slopes can be approximated with a parametric function whose single parameter can be used to characterize the degree of roughness. Facet azimuth orientation is isotropic at scales greater than 4.3 km. The simulation results indicate that surface roughness is a minor factor in modeling the sea ice brightness temperature. The change in $T_B$ is most pronounced at incidence angles greater than 40 degrees, and can reach up to 8 K for vertical polarization at 60 degrees. Therefore current and future L-band missions (SMOS, SMAP, CIMR, SMOS-HR) measuring at such angles can be affected. Comparison of the brightness temperature simulations with the SMOSice2014 radiometer data does not yield definite results.

## 1  Introduction

The L-band brightness temperature ($T_B$) is sensitive to sea ice thickness. This feature is used for sea ice thickness retrieval from L-band $T_B$ (over thin ice, <1.5 m) (Tian-Kunze et al., 2014; Huntemann et al., 2014; Kaleschke et al., 2016). Several

factors influence $T_B$ measurements over ice-covered regions, among them: ice concentration, ice temperature, snow cover, sea ice surface roughness and the shape of the interfaces between the snow and ice layers (Maaß et al., 2013; Ulaby, F. T. and Long, D. G. et al., 2014, p. 422).

Here, we investigate the effects of surface roughness on the L-band $T_B$, specifically the large scale roughness. So far this factor is not included in the modeling of sea ice emissions in operational sea ice thickness retrieval. Electromagnetic scattering

theory assumes that the roughness of a random surface is characterized by statistical parameters including standard deviation of surface height ($\sigma_z$), and correlation function ($R(\xi)$) measured in units of wavelength (Ulaby, F. T. and Long, D. G. et al., 2014, p. 422). These roughness statistical parameters are derived from measurements of surface elevation ($z$), conducted with altimeters that are characterized by their accuracy ($\delta$) and sampling distance ($\Delta x$). As usual, the measurement method has an impact on the outcome, in this case by filtering out both high and low spatial frequencies of the surface roughness. Sea ice

elevation measurements obtained from airborne altimeters (Ketchum, 1971; Dierking, 1995) and supplemented with terrestrial laser scanners (Landy et al., 2015) draw a picture of sea ice roughness as a multi-scale feature covering several orders of magnitude from large floes and pressure ridges (from tens to hundreds meters) to frost flowers and small ripples (in the centimeter to millimeter scale). The incident radiation of a given wavelength ($\lambda$) reacts differently with individual components of the superimposed roughness of many scales (Ulaby, F. T. and Long, D. G. et al., 2014, p. 252).

At small end of the roughness spectrum i.e. when the change of surface elevation over sampling distance ($\Delta z/\Delta x$) is much smaller than $\lambda$ ($\Delta z/\Delta x << \lambda$), the surface roughness is negligible. It means that the angular characteristics of scattered radiation are the same as for the secular surface. As a rule of thumb, $\Delta x$ should be smaller than $0.1\lambda$ (Dierking, 2000). Sea ice roughness measurements with terrestrial lidar carried out by Landy et al. (2015) show that standard deviation of surface height $\sigma_z$ ranges from $0.001$ m to $0.0064$ m, after high-pass filtering (cut off at $4\,\mathrm{m}^{-1}$, $\Delta x = 0.002$ m). These results indicate

that, according to the Fraunhofer's smoothness criterion ($\sigma_z < \lambda/32\cos\theta$, where $\theta$ is the angle of incidence), most sea ice types (except artificially grown frost flowers) can be treated as a smooth surface for L-band at scales lower than $0.25$ m.

In this study, we focus on the other side of the roughness spectrum i.e. the large-scale surface roughness of sea ice ($\Delta z/\Delta x >> \lambda$). In this case, changes in surface elevation are not negligible and alter the local incidence angle ($\theta_i$). Studies of surface scattering by Lawrence et al. (2011, 2013) conclude that region of $8\lambda \times 8\lambda$ is sufficient to model small-scale

roughness. Here, we assume that at larger spatial scales (larger then $8\lambda \times 8\lambda$) the surface roughness can be characterized in terms of geometrical optics (GO); for sea ice with $\epsilon_{ice} = 4.1$, $\lambda_{ice} = \lambda/\sqrt{\epsilon_{ice}} \approx 0.1$ m.

GO approximation describes the surface as a set of facets (Ulaby, F. T. and Long, D. G. et al., 2014, p. 564-567). This approach was applied for modeling the effective emissivities of mountainous terrain (Matzler and Standley, 2000) and ocean

surface (Prigent and Abba, 1990). The latter study involved probability distribution of slopes in crosswind and downwind directions. A similar method was used in the context of sea ice to assess the uncertainties caused by roughness in sea ice concentration products derived from passive microwaves (Stroeve et al., 2006). Liu et al. (2014) measured ice surface slopes and other roughness statistics in the Bohai Sea. Their result was obtained with linear (1-D) scans under the assumption of
isotropic roughness characteristics. The study by Beckers et al. (2015) has demonstrated that the statistics of sea ice roughness (mean $z$, $\sigma_z$, kurtosis and skewness) obtained from 1-D altimeter and 2-D laser scanner converge, on the condition that the surface is not strongly heterogeneous. Nonetheless, the 1-D altimeter data cannot properly represent the spatial orientation of surface facets. The surface facet orientation is characterized by both the slope ($\alpha$) and the azimuthal angle in which it is facing ($\gamma$).

In this paper, we address the knowledge gap regarding the influence of large-scale surface roughness on L-band $T_B$. The paper comprises four main sections. The introduction, presenting the context is in Section 1.

     Section 2 introduces the experimental data collected during SMOSice2014 campaign (section 2.1). Among them are the EMIRAD2 L-band radiometer $T_B$ measurements, which will serve as reference for the $T_B$ simulations. Another instrument is the airborne laser scanner (ALS) used for surface elevation measurements. The surface elevation measurements are used
to construct a digital elevation model (DEM) of sea ice surface. From the DEM we derive the facet surface slopes and their orientation. In section 2.2 we analyze the statistics of the facet orientation. Based on facet orientation statistics, we derive a parametrization of the probability distribution function of surface slopes ($PDF_\alpha$), that will serve as surface roughness representation in $T_B$ simulations.

     Section 2.3 presents the setup used for sea ice $T_B$ simulation. For the simulation of the sea ice $T_B$ we use the MIcrowave
L-band LAyered Sea ice emission model (MILLAS) (Maaß et al., 2013). In section 2.4 we show how we integrate the surface roughness statistics ($PDF_\alpha$) with MILLAS using geometrical optics.

     The three key results of this study, namely (a) surface roughness reduces the polarization difference, this change is most pronounced at incidence angles greater than 50°, (b) nadir $T_B$ is little affected and (c) comparison with the radiometer data and sensitivity study indicate that snow cover has greater impact on the $T_B$ than surface roughness are subsequently presented and
discussed in section 3.

     Section 4 summarizes the results and discusses the implications of this study for current and future L-band missions.

## 2   Materials and Methods

In this section we present the SMOSice2014 campaign that is the key dataset of this study (subsection 2.1). Section 2.2 presents the sea ice surface roughness measurements in the context of geometrical optics. Section 2.3 presents the sea ice emission model
that we used.

## 2.1 SMOSice2014 Campaign

The SMOSice2014 campaign took place between March 21, 2014 and March 27, 2014 in the area between Edgeøya and Kong Karls Land, east of Svalbard. Hendricks et al. (2014) and Kaleschke et al. (2016) described the campaign extensively. In this study we analyzed the data acquired during the flights on March 24/26. From this point onwards, we focus solely on the parts relevant to the presented work.

In the period preceding the experiment from late January until early March the meteorological conditions in the region deviated strongly from the climatological means. The air temperature measured at Hopen Island meteorological station was on average 9 to 12°C higher than the climatological value for the period 1961-1990 (Strübing and Schwarz, 2014). Prevailing southerly winds pushed sea ice against the coasts of Nordaustlandet and into Hinlopen Strait, leaving a small strip of compacted ice along the coasts of Edgeøya. When sea ice returned with southerly drift in early March, the scene was set for the experiment. The thickest, most deformed ice was located in the western part of the studied region with a gradual decrease in thickness eastwards, where thin newly-formed ice was dominant. This pattern can be observed in the SMOS sea ice thickness product displayed in Figure 1a. In this work, we focus on the data from the low altitude flight at $70\,\mathrm{m}$, because it is the data with highest spatial resolution of the Airborne Laser Scanner (ALS) among all the flights. The analysis was further reduced to the 24th of March, it is due to the fact that the region covered on the 26th of March had a discontinuous ice cover and a large scale swell was interfering with the surface elevation measurements. On March 24, the Polar 5 research aircraft of the Alfred Wegener Institute (Bremerhaven, Germany), undertook measurement flights starting from the eastern coast of Edgeøya, along the lines marked in red and green on Figure 1b. The figure also shows TerraSAR wide swath scenes, taken in the region. Flight A made between 10:05 and 10:41 UTC, flight B occured from 11:25 to 12:07 UTC. The set of instruments mounted on the aircraft included an aerial camera to visually register the ice conditions, the Heitronics KT19.85 pyrometer for surface temperature measurements, the L-band radiometer EMIRAD-2, and the Airborne Laser Scanner (ALS) for high-resolution surface elevation measurements.

### 2.1.1 EMIRAD-2 Radiometer

The EMIRAD-2 L-band radiometer (developed by DTUSpace) is a fully polarimetric system with advanced radio frequency interference (RFI) detection features (Søbjaerg et al., 2013). The setup mounted on the aircraft consists of two Potter horn antennae, one nadir pointing, one side looking at 45°incidence angle. As the aircraft flew at the altitude of $70\,\mathrm{m}$, the footprint of each antenna was $60\,\mathrm{m} \times 90\,\mathrm{m}$ for the nadir pointing antenna, and $70\,\mathrm{m} \times 90\,\mathrm{m}$ for the side looking antenna. The receiver's sensitivity is $0.1\,\mathrm{K}$ for a $1\,\mathrm{s}$ integration window. Along with the L-band measurements, navigation data was collected as to transform the polarimetric brightness temperature into the Earth reference frame (Hendricks et al., 2014). The EMIRAD-2 data was screened by evaluating kurtosis, polarimetric (Balling et al., 2012), and brightness temperature ($T_B$) anomalies. The screening showed RFI contamination for up to 30% of the samples. After subtracting of the mean value of the RFI-flagged data from the mean value of the full data, we found a $10\,\mathrm{K}$ difference for the nadir-looking horn, while the difference was non significant for the side looking horn. The analysis of the $T_B$ during the "wing wags" calibration maneuvers further revealed a

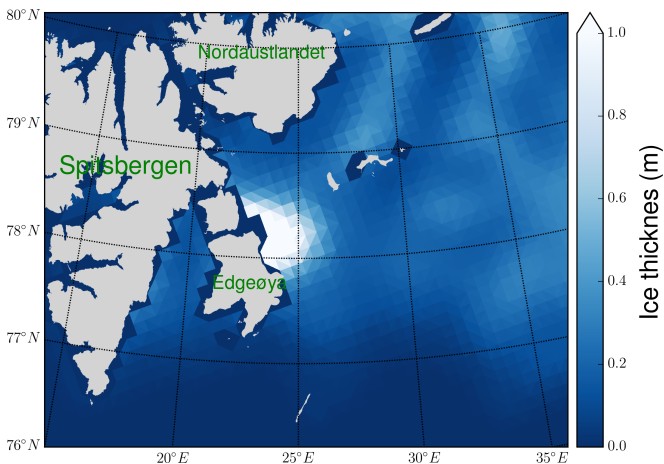

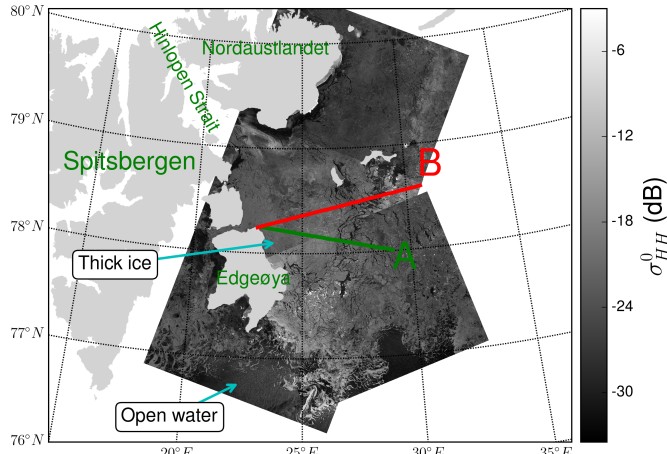

(a) Sea ice thickness on March 24 2014, derived from SMOS. The SMOS sea ice thickness product with a resolution of $40\,\text{km}$ is presented on a $15\,\text{km}$ grid. An aggregation of thick ice (>1 m) is visible along the Edgeoya's eastern coast.

(b) Sea ice conditions in the flight region on March 24. The TerraSAR-X wide swath mode (HH polarization), with frames taken at 05:35 UTC and 14:58 UTC. The aircraft tracks are marked in green - A at 10:05-10:41 UTC and red - B at 11:25-12:07 UTC.

**Figure 1.** The region of SMOSice2014 Campaign.

20 K offset relative to the nadir-looking vertical channel caused by a continuous wave signal from the camera that was mounted on the airplane to obtain visual images. The analysis concludes a purely additive characteristic that allowed for bias correction (Hendricks et al., 2014). In this study, we use the data pre-processed by the DTU-team. The radiometer data was RFI cleaned and bias-corrected and validated using aircraft wing wags and nose wags over open ocean (Hendricks et al., 2014).

### 2.1.2 Airborne Laser Scanner

In this study, the ALS (Riegel VQ-580 laser scanner) has two purposes: (1) to measure the surface elevation for subsequent estimation of the ice thickness and (2) to characterize the surface topography. The ALS near-infrared laser (wavelength $1064\,\text{nm}$) measures snow and ice elevation with the accuracy and precision of $0.0025\,\text{m}$. Across-track and along-track elevation measurements were obtained every $0.25\,\text{m}$ and $0.50\,\text{m}$, respectively. These sampling characteristics resulted from the combination of the flight altitude ($70\,\text{m}$) and the setup of the ALS (pulse repetition rate of $50\,\text{kHz}$, cross track range of $\pm 30$ degrees). The field of view of the radiometer side-looking antenna was only partially covered by the ALS scans. Nonetheless, we assume that the roughness characteristics measured by ALS are representative for both antennae fields of view. The data were calibrated and geo-referenced to the WGS84 datum. Further processing involved manual classification of tie points in leads in order to obtain local sea level and sea ice free-board (Hendricks et al., 2014). The geo-referenced surface elevations are used to compute surface roughness statistics. The elevation data is interpolated to a regular $0.5\,\text{m}$ by $0.5\,\text{m}$ grid to form a digital elevation model (DEM) of the sea ice surface. The DEM serves to derive surface slopes orientation.

The estimate of sea ice thickness was built on the hydro-static equilibrium assumption. The data required to estimate sea ice thickness consists of (a) the densities of water and ice, and (b) the snow load classically described by snow density and snow thickness, (c) ALS's free-board data.

The water, ice, and snow densities retained are $1027\,\mathrm{kg/m^3}$ (water) and $917\,\mathrm{kg/m^3}$ after Ricker et al. (2014), and $300\,\mathrm{kg/m^3}$ after Warren et al. (1999).

Snow thickness was meant to be provided by the onboard snow radar, however the equipment was still in test phase at the time of the experiment. As a workaround, we followed Kaleschke et al. (2016) and used the approximations found in Yu and Rothrock (1996) and Mäkynen et al. (2013)) that set the snow thickness to 10% of the sea ice thickness.

## 2.2 Sea Ice Surface Roughness

In this subsection we will analyze the data from the airborne laser scanner (ALS) that we presented in section 2.1.2. We use the ALS data to measure the decimeter-scale surface roughness. The ALS is a laser instrument that measures the distance to the surface. If snow covers the ice the ALS will register the snow-air interface as the elevation. Therefore, the relief of the ice is modified by snow cover. During the SMOSice2014 campaign the snow measurements were unavailable. We assume that snow cover is a plane-parallel layer over sea ice. This assumption does not account for snow dunes and drifts that may form on the ice. The implications of snow thickness on the radioactive transfer modeling are discussed in the section presenting the sensitivity analysis (section 3.2).

In the context of radiation transfer, the surface roughness is characterized in relation to incident wavelength. The ALS along-track spatial sampling of $0.5\,\mathrm{m}$ is a few times larger than the L-band wavelength in sea ice ($\lambda_{ice} = \lambda/\sqrt{\epsilon_{ice}} \approx 0.1\,\mathrm{m}$), which makes it suitable to measure the large-scale roughness, the part of the roughness spectrum where GO can be used to approximate the path of radiation. The analysis of ALS elevation data is done in three steps.

In the first step, we identify the ice with different degree of surface roughness. For that purpose we divide the flight tracks into one-second sections (approximately $70\,\mathrm{m}$ long), large enough to cover the entire nadir radiometer footprint and we build a histogram of the standard deviations of surface heights computed for these sections. The number of bins in the histogram is set according to the formula: $N_{bins} = 5\log_{10}(N_{data})$, after Panofsky and Brier (1958). We chose standard deviation as a criterion for defining the roughness classes, as it is widely used to characterize surface roughness from elevation profiles. Also, unlike visual interpretation of the aerial photography of sea ice it does not introduce personal biases. The resulting histogram in Figure 2 shows the sea ice roughness classes as histograms bins. No sections within the lowest standard deviation of surface height were found. That is probably due to the fact that no refrozen lead of the scale of $70\,\mathrm{m}$ was found or the ALS laser signal was not reflected back from the surface resulting in missing data.

In the second step, we interpolate the ALS elevation measurements to a regular $0.5\,\mathrm{m}$ grid in order to form a digital elevation model (DEM) of the sea ice surface. The sea ice surface in the DEM is represented as a set of triangular facets. Each facet orientation in the 3D Cartesian space (for simplicity we assume the base vectors $\hat{x}, \hat{y}, \hat{z}$ to be aligned with the aircraft principal

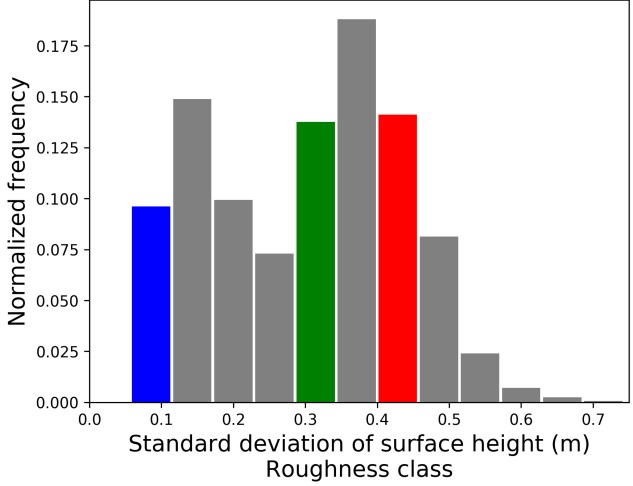

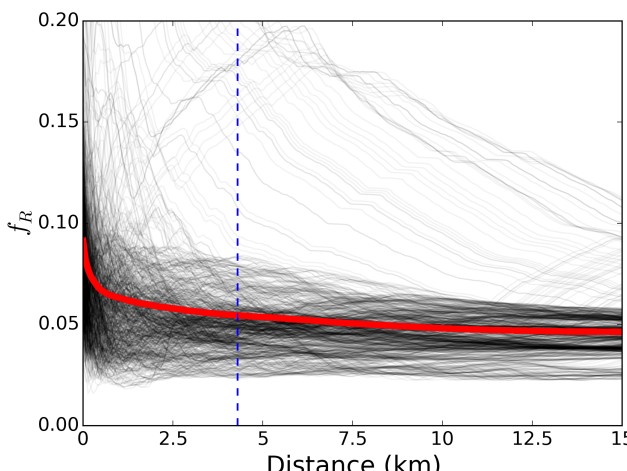

**Figure 2.** Histogram of the standard deviation of surface heights computed from 70 m flight strips, bins define the roughness classes of sea ice. Examples for three roughness classes "smooth", "medium rough" and "rough" are marked in colors blue, green and red, respectively.

**Figure 3.** The $f_R$ parameter illustrating the deviation from the uniform azimuth distribution along 1000 randomly selected samples. The average value of $f_R$ is marked as thick red line. The considered threshold of uniform distribution 4.3 km is marked by blue dashed line.

axis, so $\hat{y}$ points to the flight direction), is described by two angles: facet slope $\alpha$ ($0 \leq \alpha < \pi$) and facet azimuthal direction $\gamma$ ($-\pi \leq \gamma < \pi$). Therefore, the *i*-th facet local normal vector is described by:

$$\hat{n}_i = -\hat{x}\sin(\alpha_i)\cos(\gamma_i) - \hat{y}\sin(\gamma_i)\sin(\alpha_i) + \hat{z}\cos(\alpha_i) \tag{1}$$

In the third step, we compute the normal vectors and their orientations for the individual facets. This is done for all roughness classes. We found that the azimuthal orientation angle $\gamma$ does not show any preferred directions within any given roughness class. In the next subsections we present the analysis of the two angles characterizing the facet: azimuthal direction and surface slope.

### 2.2.1 Facet azimuth orientation

In the previous section we used the DEM to calculate the vectors normal to the surface facets. In this subsection we analyse the orientations of facet azimuths. In order to evaluate the distribution of the facet azimuths, we define parameter $f_R$ (eq. 2). This parameter is calculated from a histogram of azimuth orientation. In eq. 2 the denominator is the total number of measurments expressed as the number of angular bins multiplied by the mean number of samples per bin (aka. total number of samples). The numerator is a sum of the differences between the mean number of samples per bin and the actual number of samples in each bin. There are 36 bins. The number of bins was determined with $N_{bins} = 5\log_{10}(N_{data})$ considering the maximal

number of samples in $25\,\mathrm{km}$ flight track. The $f_R$ parameter equals to zero for the perfectly uniform distribution, in which case the number of counts in each bin ($n_i$) equals to a mean number of counts ($\mu$). The $f_R$ parameter reaches its maximum value of $f_{Rmax} = 2 - 4/N_{bins}$ when the slopes are aligned, i.e. grouped in two bins.

$$f_R = \frac{\sum_i^{N_{bins}}(|n_i - \mu|)}{N_{bins}\mu}, \qquad \mu = \frac{1}{N_{bins}} \sum_i^{N_{bins}} n_i \tag{2}$$

To evaluate $f_R$ we selected 1000 random $15\,\mathrm{km}$ samples from the flight tracks. The analysis of the samples shows that the deviation from the uniform distribution decreases sharply with increasing distance over first kilometer (figure 3). For distances along the flight track greater than $4.3\,\mathrm{km}$ the curve flattens at value of $f_R = 0.05$ in 90% of the samples. We assume that at scales greater than $4.3\,\mathrm{km}$( marked by vertical dashed line on figure 3) slope orientations do not have a preferential direction

beyond natural variability. This distance corresponds to approximately $60\,\mathrm{s}$ section. In figure 3 the average value of $f_R$ is marked as thick red line. Several sample profiles are plotted in gray lines to illustrate the variability.

### 2.2.2    Facet slope angle

Section 2.2.1 looked at the azimuthal orientation of surface facets. This section focuses on the analysis of facet slopes. For all roughness classes we observe a similar probability density function (PDF) of surface slopes. The PDFs have a maximum

at zero and a gradual decline in the likelihood of encountering the steeper slopes. Figure 4 shows the $PDF_\alpha$ in a logarithmic scale for the three distinct roughness classes: smooth $0.05\,\mathrm{m} < \sigma_z < 0.11\,\mathrm{m}$ (in blue), medium rough $0.28\,\mathrm{m} < \sigma_z < 0.34\,\mathrm{m}$ (in green) and rough $0.45\,\mathrm{m} < \sigma_z < 0.51\,\mathrm{m}$ (in red).

We decide to approximate the PDF of surface slopes with an exponential curve:

$$PDF_\alpha = C_{norm}\exp(-(\alpha/s_\alpha)), \tag{3}$$

where $C_{norm}$ is the normalization constant and $s_\alpha$ is the "geometrical-slope roughness parameter". Figure 4 presents the data and the exponential approximations. The log scale is very relevant because it becomes obvious that the chances of encountering steep slopes are getting slimmer the higher the slope angle. Consequently, it makes sense that the approximation functions misfit the observations at high slope angles as it is irrelevant to fit an approximation there. As $s_\alpha$ is the only parameter of the approximation function, it is descriptive of the surface roughness.

Figure 5 shows the relation between $s_\alpha$ and the standard deviation of surface heights corresponding to the roughness classes defined above. The error bars represent the uncertainty associated with each data point. Very rough ice has large uncertainty because the number of samples was small (classes with $\sigma_z > 0.6\,\mathrm{m}$, accounted for less than $75\,\mathrm{s}$ of flight, out of total $78\,\mathrm{min}$). The quadratic relation is holding well for ice with up to $0.5\,\mathrm{m}$ standard deviation of surface heights. The equation of the fitted curve is: $s_\alpha = 51.61\sigma_z^2 + 1.50\sigma_z + 0.14$

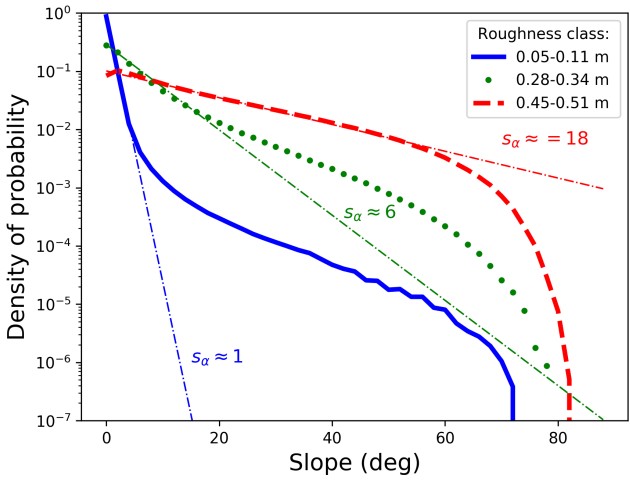

**Figure 4.** Density of probability of surface slopes in logarithmic scale for three roughness classes: smooth $0.05\,\text{m} <\sigma_z<0.11\,\text{m}$ (in blue), medium rough $0.28\,\text{m} <\sigma_z <0.34\,\text{m}$ (in green) and rough $0.45\,\text{m} <\sigma_z <0.51\,\text{m}$ (in red). The exponential fits to the respective curves are marked in thin color dotted lines.

**Figure 5.** Surface roughness parameter $s_\alpha$ describing the probability distribution of surface slopes. Error bars are inversely proportional to the number of data points in each roughness class. The "smooth", "medium rough" and "rough" classes are marked in colors blue, green and red, respectively. The red dashed line marks the fitted curve.

### 2.3 Sea Ice Brightness Temperature Simulation

In this subsection we present the emission model for simulating the sea ice brightness temperature ($T_B$). We use the MIcrowave L-band LAyered Sea ice emission model described by Maaß et al. (2013), further referred to as MILLAS. This model is based on the radiative transfer model of Burke et al. (1979) (who used it for soils), with infinite half-space of seawater covered with layers of sea ice, snow and a top semi-infinite layer of air. In contrast to the original model of Burke et al. (1979) and its usage by Maaß et al. (2013), the current version of MILLAS takes into account multiple reflections at the layer boundaries. The multiple reflections are expressed as subsequent terms of a geometric series. The summation over series stops when terms of the series contribute less than a given threshold to the total (in following calculations the threshold was set to 0.001). MILLAS describes the brightness temperature above snow-covered sea ice as a function of temperature and permittivity of the layers. The water permittivity depends mainly on the water temperature and salinity (Klein and Swift, 1977). Ice permittivity can be approximately described as a function of brine volume fraction (Vant et al., 1978), which depends on ice salinity and the densities of the ice and brine (Cox and Weeks, 1982), which in turn depends mainly on ice temperature. We set the ice salinity to $4\,\text{g/kg}$ which is a mean value for first year ice determined by Cox and Weeks (1974). The permittivity of dry snow can be estimated from its density and temperature (Tiuri et al., 1984). In the simulation, the ice and water salinity are kept constant (see Table 1). Furthermore, we assume that the system is in thermal equilibrium and that the water beneath the ice is at the freezing point. In this configuration, the $T_B$ is simulated as a function of ice thickness ($d_{ice}$), snow thickness ($d_{snow}$) and surface temperature ($T_{surf}$). In our setup, the snow is assumed to be dry with a density of $300\,\text{kg/m}^3$ , that is the average snow

**Table 1.** Brightness temperature simulation setup of the **MI**crowave **L**-band **LA**yered **S**ea ice emission model ($MILLAS$).

| | Parameter | Value |
|---|---|---|
| Snow | surface temperature | **measured** (KT19) |
| | snow wetness | 0% |
| | snow density | $300 \, \text{kg/m}^3$ |
| | snow thermal conductivity | 0.31 W/(mK) |
| Ice | ice thermal conductivity | 2.034 W/(mK) + 0.13 W/m $\cdot S_{ice}$ (g/kg)$/T_{ice}$(K) |
| | ice salinity | 4 g/kg |
| Water | water salinity | 33 g/kg |
| | water temperature | 271.2 K |

density value for December Arctic measurements from 1954-91 Warren et al. (1999). The $T_B$ simulations are only slightly sensitive to snow density, see Figure 3 in Maaß et al. (2013). The permittivities of snow and ice are linked to their temperature. The temperature profiles within snow and ice are assumed to be continuous and linear. The values for the ice and snow thermal conductivity are taken from Yu and Rothrock (1996); Untersteiner (1961). As the optimization of the emission model lies beyond the scope of this work, we use the simplest setup variant of MILLAS consisting of a single layer of ice covered with a single layer of snow.

## 2.4   Simulation of Brightness Temperature of Rough Sea Ice

In the previous sections, we described the sea ice surface as composed of facets with an orientation described by two angles: the slope $\alpha$ and the azimuthal direction $\gamma$. Subsequently, we analyzed the ALS data to extract information about statistical distributions of slopes and their orientation. We concluded that the exponential function is suitable to describe the probability density function of surface slopes for a range of ice surfaces with different degree of surface roughness.

In this section, we describe how we integrate the probability description of faceted sea ice surface with the MILLAS emission model. We will start by describing the coordinate system that we used in the $T_B$ simulations. The relations between radiometer antenna-look direction ($\hat{r}$) and the horizontal ($\hat{h}$) and vertical ($\hat{v}$) polarization vectors are described in Cartesian coordinate system. We show how these relations are represented in the coordinate system associated with a facet. The vectors defined in the tilted facet coordinate system are denoted with the subscript $i$. Subsequently, we will derive the equation that sums up the emissions originating from multiple facets.

We consider the Cartesian coordinate system ($\hat{x}$, $\hat{y}$, $\hat{z}$) with the origin in the center of the nadir sensor footprint. In this reference frame the radiometer antenna-look direction ($\hat{r}$) is described as:

$$\hat{r} = \hat{x}\sin(\theta_0)\cos(\phi_0) + \hat{y}\sin(\theta_0)\sin(\phi_0) - \hat{z}\cos(\theta_0), \tag{4}$$

where the $\theta_0$ is the antenna incidence angle and the $\phi_0$ is the azimuthal direction of the antenna. In this particular case we set the reference system so as $\phi_0 = 0$ and $\hat{x}$ is parallel to the ground. The antenna setting defines the directions of the horizontal ($\hat{h}$) and vertical ($\hat{v}$) polarization vectors:

$$\hat{h} = -\hat{x}\sin(\phi_0) + \hat{y}\cos(\phi_0), \qquad \hat{v} = -\hat{x}\cos(\theta_0)\cos(\phi_0) - \hat{y}\cos(\theta_0)\sin(\phi_0) - \hat{z}\sin(\theta_0). \tag{5}$$

We are interested in finding a relationship between the radiation originating from a tilted facet and from a flat one. For that purpose we must consider the tilted coordinate system associated with $i$-th facet (variables associated with individual facets are denoted with subscript $i$). The z-coordinate in this tilted coordinate system is aligned with the facet normal vector $\hat{n}_i$, followed by $\hat{x}_i$ and $\hat{y}_i$ calculated accordingly:

$$\hat{z}_i = \hat{n}_i \qquad \hat{y}_i = \frac{\hat{n}_i \times \hat{r}}{|\hat{n}_i \times \hat{r}|} \qquad \hat{x}_i = \hat{y}_i \times \hat{z}_i \tag{6}$$

Therefore, the local incidence angle $\theta_i$ is:

$$\theta_i = \cos^{-1}(-\hat{r} \cdot \hat{n}_i) \tag{7}$$

and the local horizontal and vertical polarization vectors are:

$$\hat{h}_i = \hat{y}_i, \qquad \hat{v}_i = -\hat{x}_i\cos(\theta_i) - \hat{z}_i\sin(\theta_i) \tag{8}$$

The emissions from the facet at an angle $\theta_i$ and polarization $p$ are denoted with an asterisk: $T_B^*(\theta_i; p)$. In order to calculate the

brightness temperatures observed in the global horizontal and vertical polarization it is necessary to account for the coordinates rotation (Ulaby, F. T. and Long, D. G. et al., 2014, p. 443):

$$T_{Bi}(\theta_i; H) = (\hat{h} \cdot \hat{h}_i)^2 T_B^*(\theta_i; H) + (\hat{h} \cdot \hat{v}_i)^2 T_B^*(\theta_i; V) \tag{9a}$$

$$T_{Bi}(\theta_i; V) = (\hat{v} \cdot \hat{h}_i)^2 T_B^*(\theta_i; H) + (\hat{v} \cdot \hat{v}_i)^2 T_B^*(\theta_i; V) \tag{9b}$$

We model the sea ice surface as a set of facets. Therefore the brightness temperature registered at the antenna aperture is a sum of contributions from N facets. We assume that each facet area $A$, at the distance $R$ is visible at the incidence angle $\theta_i$ and covers a patch of antenna field of view, equal to the solid angle $\Omega_i$:

$$\Omega_i = \frac{A\cos(\theta_i)}{R^2\cos(\alpha_i)} \tag{10}$$

The formula summing the contributions from N visible facets is:

$$T_B(\theta_0; p) = \frac{1}{N\cos(\theta_0)} \sum_{i=1}^{N} \omega_i T_{Bi}(\theta_i; p)\Omega_i \tag{11}$$

with $\omega_i$ the antenna gain component.

At this stage of our study, we aim at modeling the effect of surface roughness on the $T_B$. As a first order approximation we assume that the antenna gain is constant across the whole field of view and that the antenna is in a far field so the incidence

angle $\theta_0$ is assumed constant across the field of view. This assumption is more suitable for the space-born system. The resulting formula is:

$$T_B(\theta_0; p) = \frac{1}{N\cos(\theta_0)} \sum_{i=1}^{N} T_{Bi}(\theta_i; p)\sec(\alpha_i)(-\hat{r}\cdot\hat{n}_i) \tag{12}$$

This is the formula that we implement in the geometrical roughness model. Figure 6 presents the data flow in the geometrical roughness model. The model merges the MILLAS emission model (suitable for sea ice) with the geometrical characterization

of the faceted surface. In the presented setup the MILLAS emission model uses the sea ice surface temperature ($T_{surface}$), sea ice thickness ($d_{ice}$) and snow thickness ($d_{snow}$) as input variables. The GO part needs the cumulative probability distribution of surface slopes ($CDF_\alpha$) and the antenna look direction ($\hat{r}$). The orientation of N facets representing the sea ice surface is calculated with the inverse transform sampling (ITS) (Devroye, 2006). This method returns a random slope value from a given non-uniform distribution. The non-uniform distribution is described by a cumulative probability distribution, which in this case

depends on the geometrical roughness parameter $s_\alpha$. Similarly, the azimuthal angle $\gamma_i$ is drawn from a uniform distribution. The result of this processing step is the set of N pairs of angles ($\alpha_i$, $\gamma_i$) describing the orientation of N facets. In the next step, the local normal vector and the local incidence angle ($\theta_i$) are calculated for each of the N facets. The $\theta_i$ is used to calulate the brightness temperature emitted from the $i$-th facet with the emission model. Shadowing occurs when $\theta_i > \pi/2$ and the radiation is emitted away from the antenna. In the current setup the double-bounce effects are not accounted for. $\hat{r}$ and $\hat{n}_i$ are used to

calculate the local and global polarization directions, as well as $\Omega_i$. For the final step, the summed contributions from N facets are summed as in the equation 12. The result is the brightness temperature of the surface observed under $\theta_0$ and polarization p.

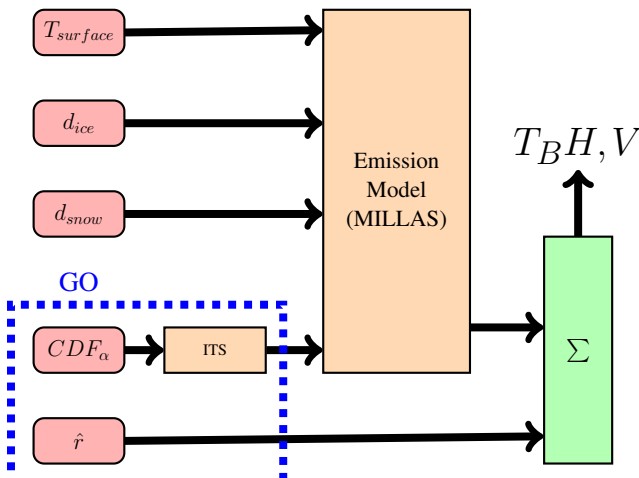

**Figure 6.** Flowchart showing the processing chain within the geometrical roughness model. The model consists of two principal blocks: the emission model and the inverse transport sampling $ITS$ module responsible for generating the facet orientation. The input parameters: surface temperature $T_{surface}$, ice thickness $d_{ice}$, snow thickness $d_{snow}$, cumulative distribution of surface slopes $CDF_\alpha$, antenna direction $\hat{r}$ are colour coded in red.

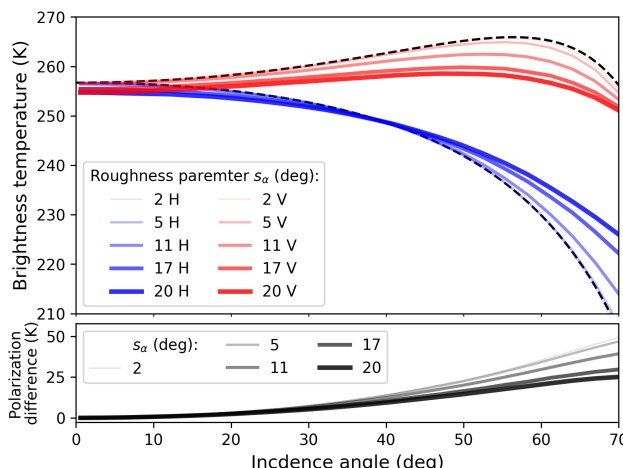

**Figure 7.** Effects of the large-scale surface roughness on the brightness temperature of sea ice, simulated with geometrical roughness model. Vertical polarization in red, horizontal in blue. The black dashed lines mark the $T_B$ for the specular surface. The surface roughness parameter $s_\alpha$ varies between 2 to 20, the thicker the line the higher the $s_\alpha$. The inputs for the MILLAS emissivity model are kept constant: $T_{surface}$=260 K, $dice$ =1.42 m, $d_{snow}$ =0.14 m.

The number of facets N impacts the quality of the simulation and the reproducibleity of the results. In order to set the value of N we looked at the standard deviation of 20 $T_B$ simulations results for nadir and 45 degrees. We decided that the standard deviation of $T_B$ should be lower than 0.1 K, as this is the accuracy of the EMIRAD-2 radiometer for the one second integration time. This criterion is met for N greater than $10^4$, which we take as the N value for further simulations.

## 3  Results

The three main results of this study are: (a) surface roughness reduces the polarization difference, this change is most pronounced at incidence angle greater than 50°, (b) nadir $T_B$ is little affected and (c) comparison with the radiometer data and sensitivity study indicate that snow cover has greater impact on the $T_B$ than surface roughness.

In section 3.1 we show the brightness temperature simulations for sea ice with different degrees of large-scale surface roughness. To interpret the results we make a sensitivity analysis (section 3.2). The comparison of the simulated vs. measured $T_B$ over 4.3 km flight track samples is shown in section 3.3.

### 3.1  Brightness temperature simulations

In section 2.2.2 we derived a parametrization of the degree of surface roughness. We approximate the roughness by an exponential probability distribution function (PDF) of surface slopes. The shape of the PDF is fully described by the $s_\alpha$ parameter.

In our simulation $s_\alpha$ varies between 2 and 20 in accordance with the measurements of surface slopes done during the SMOSice2014. As the aim of this study is to characterize the effect of surface roughness on L-band $T_B$, in this section we keep the other parameters of the emissivity model constant: surface temperature and ice and snow thickness ($T_{surface}$ =260 K, $d_{ice}$ =1.42 m, $d_{snow}$ =0.14 m). The brightness temperatures are calculated for every degree of incidence angle in range 0-70 degrees. Figure 7 shows the simulation results.

The effect of increasing surface roughness is two-fold. First, the overall near-nadir intensity is lowered by 2.6 K. Second, the polarization difference decreases. For the highest value of the roughness parameter, at Brewster's angle the vertical polarization decreases by 8 K and horizontal polarization experiences a 4 K increase. The effect of roughness is more pronounced for larger values of roughness parameter $s_\alpha$ and is most visible at higher incidence angles (60°).

The polarization mixing can be explained by the approach used in this study. The emissions from the facet in horizontal ($\widehat{h}_i$) and vertical ($\widehat{v}_i$) polarizations are partially mixed when expressed in the ($\widehat{h}, \widehat{v}$) coordinate system (see eq. 12).

The lowering of the intensity has two possible explanations. First is that our model does not take into account shadowing effects. When local incidence angle is greater than 90°, the facet is emitting away from the antenna. For the "near-nadir" angles (0°-30°), the likelihood of shadowing is less than 1% for the most rough ice (see fig. 4).

The second explanation of this effect is associated with shape of the Fresnel emissivity curves. The polarization difference for large incidence angles is larger than for the near-nadir ones. Therefore, the mean of the two polarizations ($T_B(\theta_0; H) + T_B(\theta_0; V))/2$ i.e. total intensity) is fairly constant up to 30°and than drops continuously $\sim$ 10 K by 60°. The trend continues for higher incidence angles. High values of $s_\alpha$ increase the likelihood of returning a large incidence angle in the inverse transform sampling. These large incidence angles contribute to the overall lowering of the $T_B$. The contribution of this mechanism is $\approx$2 K for $T_B(0)$ in case of the most rough ice. The 2 K estimate was obtained by integrating the drop in total intensity weighted by the $PDF_\alpha$.

The above results are obtained with a Monte Carlo simulation. This method is a time consuming approach. Therefore, we propose a parametrization of the simulation results. The two effects: the polarization mixing and the lowering of brightness temperature can be expressed in a fashion similar to the HQ model proposed by Choudhury et al. (1979). Here we propose a formulation with two parameters $H_\alpha$ and $Q_\alpha$.

$$T_B(\theta; p) = [(1 - Q_\alpha) \cdot T_B^*(\theta; p) + Q_\alpha T_B^*(\theta; q)] \cdot H_\alpha \tag{13}$$

where $p$ and $q$ stand for the polarization.

$H_\alpha$ accounts for the change in total intensity and $Q_\alpha$ for the polarization mixing. The emissions from the specular surface are denoted with an asterisk: $T_B^*(\theta; p, q)$. The proposed parametrization approximates the results obtained with the Monte Carlo simulation with a root mean square difference of $0.45\,\mathrm{K}$

$$H_\alpha = a_1 s_\alpha^2 + a_0 \qquad\qquad Q_\alpha = b_1 s_\alpha^2 + b_0 \qquad\qquad (14)$$

where $a_1 = 0.018 \times 10^{-3}$, $a_0 = 1$ and $b_1 = 0.532 \times 10^{-3}$, $b_0 = 0$.

The emissions from the specular surface are an essential input for the geometrical roughness model used in this study. The exact shape of the simulated brightness temperature curves depends on the probability distribution of slopes as well as on the emission characteristics of the specular surface. In this paragraph, we will investigate how the shape of the $T_B^*(\theta; p, q)$ influences the geometrical roughness model results. The shapes of the polarization curves i.e. the reflectivities for a given

incidence angle, are described by the Fresnel equations. Equations that are determined by the permittivity of the medium ($\epsilon$). (In this section we omit the question of penetration depth assuming that the emissions are coming from the isothermal ice layer of constant permittivity). To investigate the impact of the varying $\epsilon$ we take a range of permittivities specific to sea ice as calculated within the MILLAS model. In the present setup the sea ice permittivity depends on bulk ice salinity and ice temperature. We calculate the $\epsilon$ for a range of ice temperatures ($250\,\mathrm{K} < T_{ice} < 271\,\mathrm{K}$) and salinities ($1\,\mathrm{g/kg} < S_{ice} < 12\,\mathrm{g/kg}$).

The sea ice permitivities from the MILLAS model range between $\epsilon = 3.1 + 0.05\mathrm{i}$ (for $T_{ice} = 271\,\mathrm{K}$, $S_{ice} = 7\,\mathrm{g/kg}$) and $\epsilon = 4.6 + 0.8\mathrm{i}$ (for $T_{ice} = 253\,\mathrm{K}$, $S_{ice} = 1\,\mathrm{g/kg}$), where $T_{ice}$ is the bulk ice temperature and $S_{ice}$ is the bulk ice salinity. The curves corresponding to those values lie close together indicating that the proposed parametrization is suitable for all types of sea ice. The effect of permittivity on the polarization mixing parameter ($Q_\alpha$) is less pronounced. The dependence of the $Q_\alpha$ parameter on the roughness follows a similar quadratic curve regardless of the surface permittivity.

## 3.2   Sensitivity analysis

In this section, we investigate how sensitive the model is to its main variables: surface temperature, ice thickness, snow thickness, surface roughness and the implicit assumption of 100% sea ice concentration. This is a mandatory step toward the evaluation of the model (section 3.3).

The two most important factors influencing the L-band brightness temperature over sea ice are the ice concentration and the

ice thickness. We calculate the sensitivity of our model with regard to sea ice concentration by assuming a linear mixing of water and thick ice fractions within the radiometer footprint. The brightness temperature of sea water $T_{Bw}$ (salinity of $33\,\mathrm{g/kg}$, temperature $271.2\,\mathrm{K}$) is approximately $110\,\mathrm{K}$ and $T_{Bi}(0; p, q)$ of thick sea ice ($T_{surf} = 260\,\mathrm{K}$, $d_{ice} = 1.5\,\mathrm{m}$, bulk salinity of $3\,\mathrm{g/kg}$) is $240\,\mathrm{K}$. The resulting sensitivity to sea ice concentration is $\approx 1.5\,\mathrm{K/\%}$.

The sensitivity of the $T_B$ to sea ice thickness over thin sea ice ($d_{ice} < 0.75\,\mathrm{m}$) is fundamental for the sea ice thickness

retrieval from L-band radiometry. $T_{Bi}$ saturates when the sea ice thickness is significantly larger than the penetration depth of

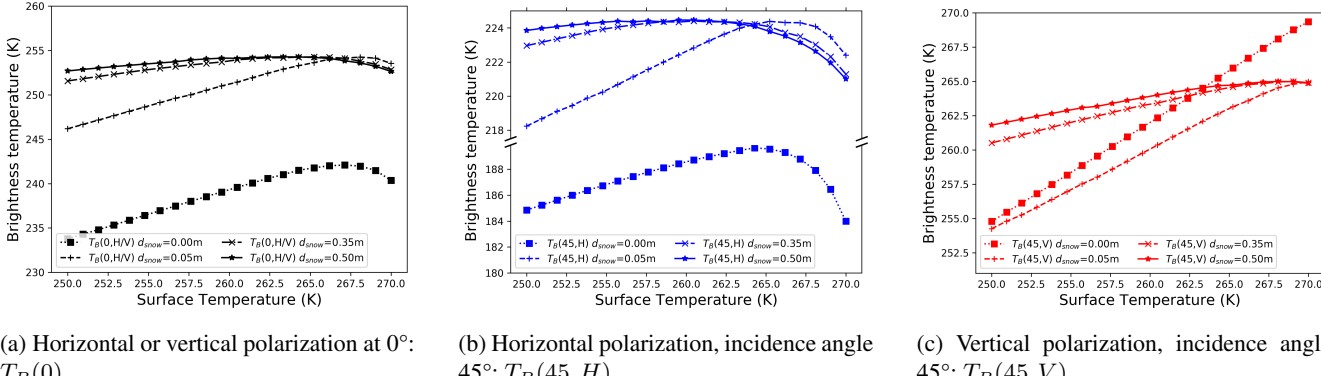

(a) Horizontal or vertical polarization at 0°: $T_B(0)$

(b) Horizontal polarization, incidence angle 45°: $T_B(45, H)$

(c) Vertical polarization, incidence angle 45°: $T_B(45, V)$

**Figure 8.** Change in the brightness temperature as predicted by the MILLAS emission model as a function of surface temperature. The different line styles correspond to the different snow thickness assumptions. The calculation was done for the sea ice thickness of one meter and surface roughness set to zero.

the L-band radiation. Therefore, our analysis focuses on sea ice thicker than $1\,\text{m}$, in order to single out the contributions of surface roughness.

Table 2 contains the sensitivities of the geometrical roughness model to the input parameters: roughness parameter $s_\alpha$, ice thickness $d_{ice}$, snow thickness $d_{snow}$ and surface temperature $T_{surf}$. Presented values are grouped into columns corresponding

to the polarization and three incidence angles: 0°, 45°and 60°. The angles are chosen to reflect the antennae configuration during the SMOSice2014 with an addition of 60°, close to Brewster's angle where surface roughness effects are most pronounced.

The assumption about snow thickness has a considerable effect on the sea ice $T_B$ (Maaß et al., 2013). For this reason the values of sensitivities are considered for a number of snow thickness values. Figures 8a, 8b and 8c show the simulated L-band brightness temperature at 0°, 45°as a function of the surface temperature. In this study we make considerable assumptions

about snow thickness. To illustrate the uncertainty associated with the assumptions, the simulations are made for a range of snow thicknesses plotted in different line styles.

In the MILLAS model, ice permittivity is parameterized with ice temperature. The non-monotonic shape of the curves is caused by change in ice permittivity. Therefore, the sensitivities are calculated for two temperature ranges: lower ($250\,\text{K}$-$265\,\text{K}$) and higher ($265\,\text{K}$-$270\,\text{K}$). Table 2 contains the calculated sensitivities.

As far as the large-scale surface roughness is concerned, the sensitivity increases almost linearly for the values of $s_\alpha$ between 0°and 20°which is the maximal value measured during the SMOSice2014 campaign.

In order to interpret the results of the comparison between the simulation and measurements, it is necessary to evaluate the uncertainties associated with the input parameters: surface temperature, ice thickness and snow thickness. In the following paragraphs, we use "mean model sensitivity for the cold conditions" for the averaged absolute sensitivity of $T_B(0; H, V)$ and

$T_B(45; H, V)$ at $250\,\text{K}$. We take the values for the lower temperature range as they match the conditions during SMOSice2014 campaign.

**Table 2.** Table with sensitivities of the brightness temperature at nadir, 45°and 60°as simulated with Geometrical Optics surface roughness model. The input parameters: roughness parameter $s_{alpha}$, ice thickness $d_{ice}$, snow thickness $d_{snow}$, surface temperature $T_{surf}$.

| | $T_B(0;H,V)(K)$ | | $T_B(45;H)(K)$ | | $T_B(45;V)(K)$ | | $T_B(60;H)(K)$ | | $T_B(60;V)(K)$ | |
|---|---|---|---|---|---|---|---|---|---|---|
| $\partial/\partial s_\alpha$ \| $d_{snow}$=0 | -0.01 to -0.20 | | 0.06 to 0.80 | | -0.08 to -1.20 | | 0.11 to 1.70 | | -0.14 to -2.10 | |
| $\partial/\partial s_\alpha$ \| $d_{snow}$=0.1m | -0.01 to -0.21 | | 0.02 to 0.30 | | -0.05 to -0.74 | | 0.05 to 0.77 | | -0.08 to -1.19 | |
| $\partial/\partial s_\alpha$ \| $d_{snow}$=0.25m | -0.01 to -0.21 | | 0.02 to 0.33 | | -0.05 to -0.76 | | 0.05 to 0.80 | | -0.14 to -1.22 | |
| $\partial/\partial s_\alpha$ \| $d_{snow}$=0.35m | -0.01 to -0.21 | | 0.02 to 0.34 | | -0.05 to -0.77 | | 0.05 to 0.82 | | -0.14 to -1.24 | |
| $\partial/\partial s_\alpha$ \| $d_{snow}$=0.45m | -0.01 to -0.21 | | 0.02 to 0.35 | | -0.05 to -0.78 | | 0.05 to 0.83 | | -0.14 to -1.25 | |
| $\partial/\partial s_\alpha$ \| $d_{snow}$=0.5m | -0.01 to -0.21 | | 0.02 to 0.37 | | -0.05 to -0.79 | | 0.05 to 0.85 | | -0.14 to -1.26 | |
| range (K) | 250-265 | 265-270 | 250-265 | 265-270 | 250-265 | 265-270 | 250-265 | 265-270 | 250-265 | 265-270 |
| $\partial/\partial d_{ice}$ | -0.91 | 2.05 | 0.18 | 3.49 | -1.77 | 0.16 | 0.58 | 3.28 | -2.12 | -1.24 |
| $\partial/\partial d_{snow}$ | 8.51 | -2.03 | 6.58 | -3.18 | 10.71 | 0.23 | 5.68 | -2.45 | 10.10 | 0.89 |
| $\partial/\partial T_{surf}$ \| $d_{snow}$=0 | 0.50 | -0.43 | 0.4 | -0.95 | 0.61 | 0.15 | 0.29 | -1.32 | 0.69 | 0.66 |
| $\partial/\partial T_{surf}$ \| $d_{snow}$=0.1m | 0.34 | -0.23 | 0.3 | -0.5 | 0.41 | 0.02 | 0.26 | -0.63 | 0.44 | 0.19 |
| $\partial/\partial T_{surf}$ \| $d_{snow}$=0.25m | 0.18 | -0.38 | 0.12 | -0.59 | 0.24 | -0.11 | 0.09 | -0.68 | 0.27 | 0.05 |
| $\partial/\partial T_{surf}$ \| $d_{snow}$=0.35m | 0.11 | -0.39 | 0.05 | -0.61 | 0.17 | -0.16 | 0.01 | -0.66 | 0.21 | -0.01 |
| $\partial/\partial T_{surf}$ \| $d_{snow}$=0.45m | 0.06 | -0.40 | 0.01 | -0.58 | 0.13 | -0.19 | -0.03 | -0.67 | 0.17 | -0.05 |
| $\partial/\partial T_{surf}$ \| $d_{snow}$=0.5m | 0.05 | -0.40 | 0.01 | -0.57 | 0.11 | -0.19 | -0.05 | -0.66 | 0.15 | -0.06 |

The uncertainty associated with surface temperature is estimated as the product of the sensitivity of the model times the measurement uncertainty. The surface temperature measurements done with the KT19.85 have an accuracy of $0.5\,\mathrm{K}$. The mean surface temperature in the region covered by ice was $251.7\pm3.5\,\mathrm{K}$. We take the standard deviation of surface temperature measurements as the parameter uncertainty. We estimate the uncertainty associated with surface temperature to be $0.7\,\mathrm{K}$.

5     The uncertainty associated with sea ice thickness is estimated as the product of the sensitivity of the model times the measurement uncertainty. The sea ice thickness measurements in this study are derived from the re-sampled $ALS$ elevation data. The mean standard deviation of the re-sampled elevation measurements is $0.08\,\mathrm{m}$. The assumption about the densities of snow, ice and water combined with the assumption on the snow thickness of 1/10 of ice thickness are leading to the uncertainty of $0.4\,\mathrm{m}$. Taking into account the mean model sensitivity for the cold conditions prevailing during the flights, we estimate that

10   the uncertainty associated with sea ice thickness is $0.5\,\mathrm{K}$.

The uncertainty associated with snow thickness is estimated as the product of the sensitivity of the model times the measurement uncertainty. Unfortunately, the snow thickness measurements are not available. The snow layer, although transparent for the L-band radiation, is not invisible. The refraction on the snow-ice and snow-air interfaces alters the local incidence angles. Snow cover also has an effect on the temperature profile within the ice. This indirectly affects the permittivity of sea ice. All these factors make an estimation of the uncertainty caused by snow thickness especially hard to quantify. We assume that snow thickness uncertainty is equal to the mean standard deviation of the re-sampled elevation measurements, that is $0.08\,\mathrm{m}$. The mean model sensitivity to snow thickness for the cold conditions is $8.6\,\mathrm{K/m}$. Therefore, we estimate the uncertainty associated with snow thickness to be $0.7\,\mathrm{K}$.

An important factor which is not directly included in the model is the sea ice concentration. In the model we assume sea ice concentration to be 100% in order to single out the much smaller contribution of surface roughness. However, if a linear mixing model is applied the sensitivity to sea ice concentration is $-1.5\,\mathrm{K/\%}$. During the pre-processing of the airborne laser scanner (ALS) data we excluded the seventy-meter sections with more than 5% missing values. The missing values are caused by the instrument setup (rotating mirror, edge of swath) or by the lack of return reflection from open water or thin ice. We estimate that the uncertainty associated with the sea ice concentration is up to $7.5\,\mathrm{K}$.

To put the partial sensitivities into perspective, the expected changes in the $T_B$ caused by the strongest surface roughness measured during SMOSice2014 campaign do not exceed $-2.2\,\mathrm{K}$ for nadir and $1\,\mathrm{K}$ and $-5.6\,\mathrm{K}$ for the horizontal and vertical polarization of the 45°antenna, respectively.

To conclude, the sensitivity analysis of the geometrical roughness model leads to the conclusion that the surface roughness effects will be hard to observe in the SMOSice2014 flights data with the current emission model setup.

## 3.3  Simulations vs. Measurements

In this section, we compare the brightness temperature measured with the EMIRAD-2 radiometer with the brightness temperature simulations. The comparison is done on a $4.3\,\mathrm{km}$ section as to justify the assumption of the isotropic azimuth distribution. We want to determine the simulation setup that best reproduces the radiometer measurements. And whether the inclusion of the surface roughness in the simulation brings significant improvement. The limitation of this approach is that we assume that the ice observed by the side looking antenna and the ice below the flight path have the same properties. We consider the surface temperature, the sea ice thickness and the surface roughness along the flight and we use them to run the statistical roughness model, described in section 2.4 with MILLAS single ice layer setup as the brightness temperature module. In setups including snow layer, the snow thickness is set to be 10% of the sea ice thickness. The calculation is done for $60\,\mathrm{s}$ averages, during which the aircraft covers the distance of approximately $4.3\,\mathrm{km}$. At this scale we consider the surface slopes orientation to be isotropic. The measured surface slopes are used to compute empirical $CDF_\alpha$. Analysis at $4.3\,\mathrm{km}$ scale is justified by the fact that the antennae gains are not known. Furthermore, the averaged sea ice properties are more representative for the space-born radiometers.

For each radiometer channel we made four simulation setups: two without roughness (Flat no- snow, Flat snow), and two with roughness included: (Rough (GO) no-snow, Rough (GO) snow). As for the performance metrics of the model setups, we

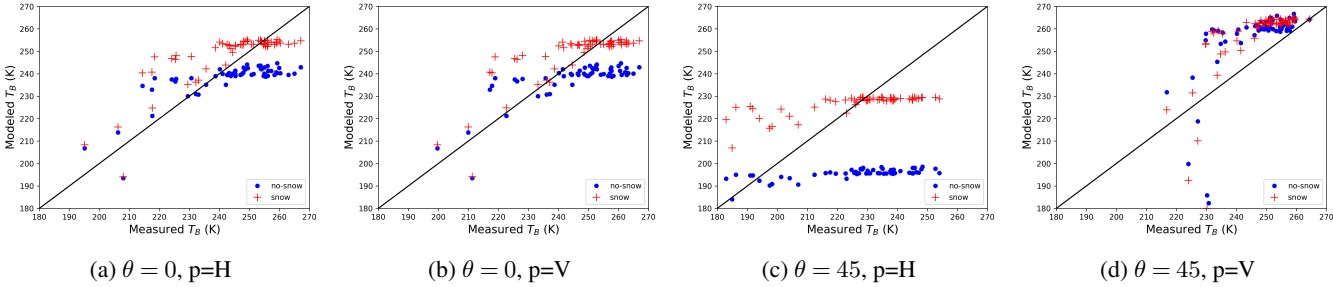

**Figure 9.** Scatter plots illustrating the comparisons between the EMIRAD-2 data and the $T_B$ simulated without GO roughness included - **specular**. Results corresponding to the setup with snow are marked in green, without snow in blue.

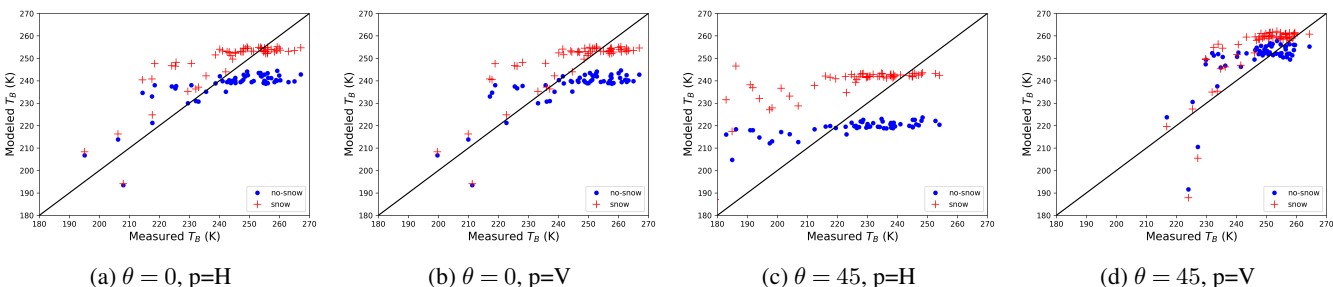

**Figure 10.** Scatter plots illustrating the comparisons between the EMIRAD-2 data and the $T_B$ simulated **with GO roughness included**. Results corresponding to the setup with snow are marked in green, without snow in blue.

use the coefficient of determination ($r^2$), the root-mean-square error (RMSE) and bias. These metrics are widely used in the assessment of the performance of satellite measurements (Entekhabi et al., 2010). Table 3 contains the results of the comparison expressed in terms of $r^2$, RMSE and bias. The corresponding scatter-plots illustrating the comparison between measured and modeled brightness temperatures are presented on figures 9 and 10.

The values of $r^2$ for all "channel - simulation setup" combinations do not exceed 0.36. The simplified one-layer model managed to capture only 36% of the signal variance even with surface roughness included. Furthermore, the inclusion of surface roughness brings little improvement to the statistics. In case of vertical polarization, where the model studies indicate the most sensitivity to roughness, the $r^2$ is even a little lower. The inclusion of a very crude snow thickness parametrization is more successful in capturing the radiometer measurements variability. All metrics show that the four model setups perform

poorly in reproducing the EMIRAD-2 measurements.

    The results of the comparison are also presented in the form of histograms of the difference between the measured and simulated $T_B$ (figure 11). For all four antenna feeds the RMSE between simulated and measured $T_B$ decreases whenever the setups include snow. However, the bias gets higher.

    The high values of RMSE and bias show a general miss-fit of the model to the data. The sensitivity study of the model

presented in section 3.2 indicates that the effects of surface roughness are of comparable magnitude or smaller than uncertainties associated with the model input parameters. The side looking vertical polarization is predicted to be most affected by the

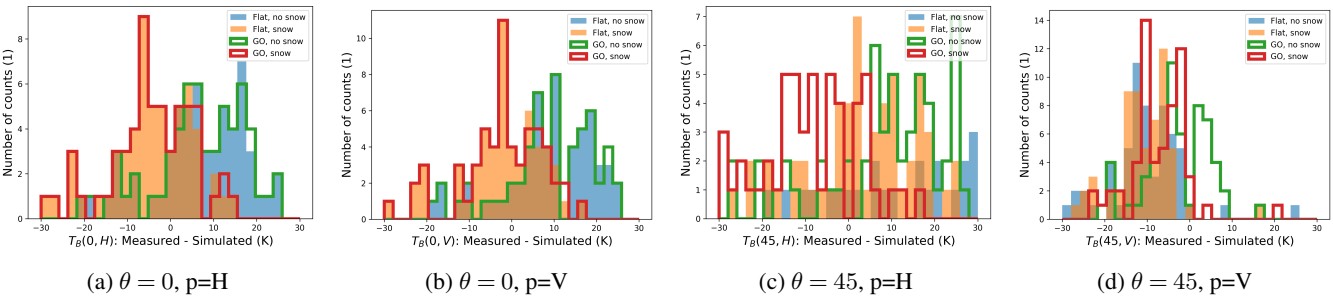

| (a) $\theta = 0$, p=H | (b) $\theta = 0$, p=V | (c) $\theta = 45$, p=H | (d) $\theta = 45$, p=V |

**Figure 11.** Histograms of the differences between the EMIARAD2 measurements and simulation setups for four antenna feeds.

surface roughness. However, the surface roughness is not measured directly in side-looking antenna footprint. The scatter plots on figures 9 and 10 and the calculated statistics show that a simplified one-layer model struggles to capture the dynamics of the sea ice $T_B$ regardless of the simulation setup.

**Table 3.** Performance metrix for the different $T_B$ simulation setups in terms of coefficient of determination r², RMSE (K), bias (K). For EMIRAD-2 channels four model setups are tested: Flat no snow, Flat snow, GO no snow, GO snow.

| | | $T_B(0,H)$ | | | $T_B(0,V)$ | | | $T_B(45,H)$ | | | $T_B(45,V)$ | | |
|---|---|---|---|---|---|---|---|---|---|---|---|---|---|
| | | $r^2$ | RMSE (K) | bias (K) | $r^2$ | RMSE (K) | bias (K) | $r^2$ | RMSE (K) | bias (K) | $r^2$ | RMSE (K) | bias (K) |
| Setup | Flat, no snow | 0.27 | 27.2 | 11.7 | 0.26 | 28.5 | 6.8 | 0.22 | 41.7 | 71.3 | 0.19 | 26.7 | -41.2 |
| | Flat, snow | 0.36 | 23.7 | -18.2 | 0.35 | 24.1 | 23.9 | 0.33 | 25.0 | 68.5 | 0.29 | 25.2 | -89.8 |
| | GO, no snow | 0.27 | 27.3 | 12.3 | 0.25 | 28.5 | 7.4 | 0.26 | 28.2 | 78.2 | 0.19 | 26.0 | -41.9 |
| | GO, snow | 0.36 | 23.7 | -17.8 | 0.35 | 24.1 | -23.4 | 0.34 | 27.2 | 72.2 | 0.28 | 24.7 | -108.7 |

## 4 Discussion and Conclusions

5    In this paper we address the knowledge gap concerning the influence of the decimeter-scale surface roughness on the L-band brightness temperature of sea ice. We used the airborne laser scanner (ALS) data to characterize the sea ice surface and to produce the digital elevation model (DEM) of the sea ice surface. From the DEM we derived the probability distribution of surface slopes ($\alpha$) and their azimuthal orientation ($\gamma$). We found that the probability distribution function of $\alpha$ ($PDF_\alpha$) can be described with an exponential function regardless of the degree of roughness of sea ice surface. The exponent parameter ($s_\alpha$) is

10    a quasi-quadratic function of the standard deviation of surface heights. In the second part of this work, we used the $PDF_\alpha$ in a Monte Carlo simulation of the emission from a faceted sea ice surface. The effect of surface roughness is marginal at near-nadir, accounting for up to $2.6\,\text{K}$ decreases in $T_B$. The polarization curves around Brewster's angle are most affected. The vertical polarization decreases by $8\,\text{K}$ and horizontal polarization increases by $4\,\text{K}$ for the roughest ice, compared with the specular sea

ice surface. The effect of the large-scale surface roughness on polarization curves is not linear with the degree of the surface roughness described by $s_\alpha$. Meaning that the alteration of the $T_B$ curves is strongest for the roughest surface. The overall change of emission due to the large-scale surface roughness can be expressed as a superposition of change in intensity ($H_\alpha$) and an increase in polarization mixing ($Q_\alpha$). The change in intensity depends primarily on the surface permittivity, whereas

the polarization mixing shows little dependence on $\epsilon$. This parametrization is suitable for all types of sea ice. However, the sensitivity analysis of the simplified emission model demonstrates that the expected change in $T_B$ is comparable in magnitude to the uncertainty associated with the model input parameters.

The results have implication for the current and future L-band missions. The operational SMOS sea ice thickness product relies on near-nadir $T_B$ observations (0°-30°). Therefore, the large scale surface roughness will have little effect on the re-

trieval. The SMAP and CIMR missions, that operate at incidence angles of 40°and 55°, respectively, are more exposed to the surface roughness effects. The effect on the vertical polarization is stronger than on the horizontal polarization. Lastly, we compared the simulation of the brightness temperature (with and without surface roughness) with the radiometer measurements. Unfortunately, this showed that our model is not capturing the brightness temperature variability at the scale of $4.3\,\mathrm{km}$. The inclusion of surface roughness is less important than the inclusion of a crude snow thickness parameterization. This is

confirmed by the sensitivity analysis of the model. Another possible explanation is that the sea ice in the studied region was highly heterogeneous in terms of its thickness and snow cover. Furthermore, a simple two-layer emission model used in this study has its limitations in capturing the $T_B$ variability. Better results might be obtained if a multi-layer model together with the snow thickness measurements is used. With such setup the direct inclusion of sea ice facets orientation in the radiometer field of view will be a valuable option to improve the $T_B$ simulation. However, this would require in-situ measurements of sea

ice permittivity, snow thickness, temperature and roughness as well as detailed characterization of the antenna gain. Thus, the authors recommendation for future studies is to measure the microphysical snow and sea ice properties together with surface roughness directly in the radiometer's field of view.

*Code and data availability.*  Code and data are available from the authors on request.

*Author contributions.*  Conceptualization, M.Miernecki and L.Kaleschke; Methodology, M.Miernecki; Software, M.Miernecki and N.N.Maaß;

Validation, M.Miernecki, N.Maaß; Formal Analysis, M.Miernecki and L.Kaleschke; Investigation, M.Miernecki; Resources, L.Kaleschke; Data Curation, S.Hendricks, S.S.Søbjrg; Writing—Original Draft Preparation, M.Miernecki; Writing—Review & Editing, L.Kaleschke, N.Maaß ; Visualization, M.Miernecki; Supervision, L.Kaleschke; Project Administration, L.Kaleschke; Funding Acquisition, L.Kaleschke

This research was funded by the HGF Alliance, Remote Sensing and Earth System Dynamics. The European Space Agency co-financed the AWI research aircraft Polar 5 and helicopter flights (ESA contract 4000110477/14/NL/FF/lf; PI S.Hendricks)

and the development and validation of SMOS sea ice thickness retrieval methods (ESA contracts 4000101476/10/NL/CT and

4000112022/14/I-AM; PI L.Kaleschke). Technical University of Denmark (DTU) co-financed and conducted the measurements with EMIRAD2 L-band radiometer on Polar 5.

*Competing interests.* The authors declare no conflict of interest. The founding sponsors had no role in the design of the study; in the collection, analyses, or interpretation of data; in the writing of the manuscript, and in the decision to publish the results

5   *Acknowledgements.* The authors acknowledge the institutions providing the data and people involved in carrying out the measurements. The TerraSAR-X and TanDEM-X teams provided the SAR data for this study.

Special thanks to Dr. Yann Karr, PI of the SMOS mission, for his suggestions and comments. As well as the financial support that covered the publication costs.

The following abbreviations are used in this manuscript:

| | |
|---|---|
| ALS | Airborne Laser Scanner |
| CDF | Cumulative Distribution Function |
| DEM | Digital Elevation Model |
| GO | Geometrical Optics |
| ITS | Inverse Transform Sampling |
| MILLAS | MIcrowave L-band LAyered Sea ice emission model |
| PDF | Probability Distribution Function |
| RFI | Radio Frequency Interference |
| SAR | Synthetic Aperture Radar |
| SMAP | Soil Moisture Active and Passive Mission |
| SMOS | Soil Moisture and Ocean Salinity Mission |
| $T_B$ | brightness temperature |

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
