# Peer review of "Effects of decimetre-scale surface roughness on L-band Brightness Temperature of Sea Ice"

_The Cryosphere, 2019_

## Referee Comment (RC1) · Jack Landy (Referee) · 17 Jun 2019

This study offers a novel and interesting look at the effect of surface roughness on measured sea ice brightness temperatures at L-band – an important and relatively understudied topic. The method of applying Geometrical Optics to model the sea ice emissivity from observed PDFs of the ice surface height distribution seems reasonable. Results from theoretical simulations are a valuable contribution to the sea ice remote sensing community.

However, it remains unclear from the study what the relative importance of decimetre-scale roughness is to variability in sea ice brightness temperatures, in comparison

to say ice thickness, thermodynamic state, snow depth, snow properties and open water within the radiometer footprint. I have provided a set of general comments on the methodology and recommendations for improving the analysis or taking it further. I've also made some minor suggestions to improve the readability of the paper and clarify a few confusing statements. I'd recommend this manuscript is reconsidered for publication in The Cryosphere following these revisions.

Please do get in contact if you have questions regarding these comments. Kind regards, Jack Landy

General comments:

1. It is not obvious from the paper what are the implications of your results for sea ice thickness measurements from satellite, e.g. SMOS or the upcoming CIMR mission. What is the relative importance of decimetre roughness compared to other factors? Do the current incidence angles employed by SMOS limit the sensitivity of measured Tb to roughness? I expected to see a statement on this in the abstract and some discussion later on the manuscript.

2. The airborne altimeter data provide measurements of the snow surface roughness, but this is not necessarily reflected directly in the underlying ice-snow interface roughness. There is no discussion of this in the manuscript and the potential issues/errors it could introduce. Which is most important for L-band emissivity, snow or ice surface roughness? Might the roughness be overestimated if it's the ice interface roughness that you need to know?

3. It is not clear whether the assumption of isotropically-oriented surface roughness features is valid, even when averaging model-data comparisons over 5 km. Is the sensitivity of modelled Tb to surface feature orientation linear? When modelling Tb over 5 km, the assumption is that Tb will be the average of a uniform distribution of surface feature azimuth angles. But is it reasonable to assume the average of short radiometer Tb integrations, from sea ice with lots of different surface feature orientations, is measuring the same thing? Is there any geometrical shadowing of facets at the 45-degree incidence angle? If so, how do you account for this in the simulations?

4. Could you not just use the observed empirical CDF within each 70 m footprint, rather than the statistical model fit, to simulate Tb? i.e. integrate over the N pairs of angles for each facet within the 70 m footprint. Is this just to speed up your simulations ($70^2/0.5^2$ is only about factor 2 larger number of facets than your $10^4$ criteria), or so you can calculate average model results over 5 km sections? Using the observed CDF may produce a better model fit to the radiometer observations.

5. A particularly useful contribution of this paper would be a more in-depth model sensitivity analysis of the relative effect of roughness on measured Tb compared to other factors. The current results touch on this with e.g. Fig 6, but by keeping other factors constant in the simulations its impossible for the reader to understand the true sensitivity to roughness. For example, how different does Fig 6 look for a different set of sea ice constants? E.g. 3 m thick, fresh MYI, with thicker snow depth and a warmer surface? I would recommend removing Figs 7 and 8, which don't really contribute to the message of the paper, and adding some deeper theoretical analysis of the relative impact of roughness on L-band Tb.

6. Results from the comparison between modelled and observed Tb are not promising and are difficult to interpret here. I had many questions looking at Figs 9 and 10 that were not discussed within the text. Based on the theoretical results in Fig 6, you'd probably only expect an improvement to the 45-degree angle v-pol channel when including the effects of roughness, right? So the poor correspondence between model and observations is likely one or more of: model inaccuracy, the model configuration (no. layers, penetration etc.) not being adequate, simple treatment of ice thermodynamics, not having altimeter observations for the 45-degree footprints, or the limited treatment of snow. Without showing results from a model sensitivity analysis of these factors though, it is impossible to interpret which factor or set of factors is most likely. Why is there such a low dynamic range for modelled Tb's in most cases where measured Tb >220-240K? Can you add another figure showing the absolute differences between modelled Tb for simulations with and without GO roughness included, perhaps as histograms or as a function of the surface roughness?

7. The written English needs some improvement throughout the manuscript. I would recommend a careful proof-read to check spelling and grammar. A few e.g.'s just on the first page are: L2 'rely on', L8 you mean 'horizontal polarization'?

Minor comments/edits:

Page 1. Line 7. Effect on what? What scale of roughness? Multiple scales?

L 18. Surface roughness of the ice or snow, or both?

P2 L5. Both high and low spatial frequencies. . .

L11. What do you mean by 'stays unnoticed'? rephrase

L18. What is 8*lambda for L-band?

Intro Section. What about the other factors affecting Tb from sea ice? They are not the primary focus of this study, but can complicate your interpretations of the roughness effects, particularly when comparing model results to radiometer observations. So you should introduce the effects of e.g. thermodynamics, ice concentration, snow properties etc. here. Are there any previous estimates of the impact of roughness on L-band Tbs?

P3 L8-9. What was the air temperature then? Thermodynamic effects on the snow properties may explain your difficulties comparing model and observations, and the large impact of a snow layer on your simulations then?

P4 L1-3. Can you provide a little more detail on this as its such a substantial bias? How do you know it was purely additive? How was this tested?

P4 L10. Here I found myself asking if you used the same roughness data from nadir to

simulate the 45-degree return. This was answered much later but you should state it here.

P5 L2-3. How was the sea level estimated from lead tie-points? Do you have uncertainties for the sea level, freeboard and ice thickness, that you can apply to estimate uncertainties in modelled Tb? How did you estimate snow depth uncertainty when applying a simple snow scheme?

L4. Do you use an iterative procedure to estimate ice thickness then, if the thickness is already required to estimate snow depth?

L11. Do you have a citation for the version of MILLAS used here, or was this added work completed as part of your study? If the latter, you need to describe model additions including equations for review (perhaps in an appendix).

L16. How are the ice and water salinities calculated? (You need to make it clear here that you use the ice surface T from airborne observations to constrain ice thermodynamics in the simulations). I'd like to see a table here of the constants used for ice, water and snow physical parameters, and then the range over which other parameters (e.g. roughness) varied.

L30-31. It's important here that you state ALS observations of roughness are averaged over quite a long window. As it's currently written, it sounds like you are simulating and comparing with measured returns over the exacts same 70 m window.

L34. Use a proper citation style for this reference.

P6 L4-5. Did you filter out all 70-m sections containing mixed classes, e.g. some open water? What about thin leads within the footprint? I'd expect many of your eventual 5-km sections contained at least some open water, so how was this accounted for?

L13. 'coast'

P7 L1-2. I'd like to see a figure which proves this. This cutoff limit between anisotropic

and isotropic orientation of surface features has not been shown before, so a novel result of this study. But if you want to prove there is a scale separation at 4.3 km you need to show the data

L10. Can you show the exponential function fit to each class of data in Fig 3, so we can see how well it performs?

Eq 9. What is R?

P10 L3-4. Is it reasonable to assume a constant gain pattern over the entire FOV? Do you have an estimate of the antenna pattern to compare to?

L10. So the T profile is calculated directly from surface T and the reference constant salinity?

P12 L5. If you refer to angles in degrees within the text, the x-axis in Fig 6 should also be in degrees

L9. 'And'?

L12-13. Confusing. What do you mean by this?

P13 L3. 'High'?

P13 L16-31. I can't understand why this section is included, along with Figures 7 and 8. Why not just calculate reasonable variations in MILLAS emissivity for different sea ice scenaios? E.g. warm/cold ice, different salinities, shallow/deep snow, different snow T or densities? Relative permittivities up to 10-20 are unrealistic for sea ice in most conditions, so they are not helpful for your analysis here. There's only really reason to show the cases listed directly on Line 24.

P15 L5. State this earlier in the method.

L7. How do you decide when it is needed?

L8-9. Is the roughness CDF calculated from all altimeter observations within this 5-km

window then?

P18 L8. Unlikely permittivity but possibly thickness. What about open water within footprints? Could that have affected the radiometer measurements? Or maybe snow depth/property variations along track?

L9-10. You at least had the facet orientation info at least for the nadir looking antenna right?
* * *

---

## Referee Comment (RC2) · Georg Heygster (Referee) · 18 Jul 2019

This manuscript presents a method to model the effects of decimeter-scale surface roughness on the L-band signal of sea ice, and compares the results with airborne observation. While the model indicates a clear result of brightness temperature reduction of up to 8 K (v-pol) resp. 2.6 K (v-pol), the comparison with the experiments yields little correlation of roughness and brightness temperature. Although the results are not very indicative, the manuscript treats an important subject.

Main comments:

[Figure]

The discussion of Fig. 6 and its use for interpretation of the experiments is incomplete: One of the interesting results of Fig. 6 is that between 40° and 45° inc angle, the h-pol TBs are practically insensitive to the roughness parameter s. This is important for L band satellite sensors observing only at such incidence angles like SMAP (in orbit since 2015) and the upcoming CIMR, and for the airborne observations at 45° (Figs. 9 and 10, Table 1): In the case when no influence of the roughness on the TB signal is expected (h-pol), the found correlation between observation and model is clearly higher, the RMSE, bias and the ubRMSE all are higher than at v-pol, where the model predicts a sensitivity to roughness. In Figs. 9(c) and 10(c), the h-pol 45 ° inc angle cases, the modeled TBs show clearly less variability than the corresponding v-pol cases (Figs. 9(d) and 10(d)). Do you have an interpretation for this finding?

Fig. 6 and P12 L9-10 '..the horizontal and vertical polarization curves are brought together.' Correct only at incidence angles > 45°. At lower angles, the opposite is the case. Best, add to Fig. 6 the polarization difference curves near the bottom, potentially at an increased y scale.

Fig.6: Give x axis in degree, not in rd because in text you use deg.

The current version of eq. (3) contains a product instead of a sum ('+' missing), and eq. (5) is incorrectly copied from Ulaby and Long, (2014), p443: replace ˆr in nominator and in denominator by ˆn, and check order of factors. ˆr in this equation does not make sense at all: ˆy should be independent of ˆr !

Units should be given in a consistent way throughout the whole manuscript. Here, the units m, cm and mm all occur, which is confusing and makes reading cumbersome.

Have always a blank between number and units.

The references in the text are frequently odd: if part of a sentence, then it should read 'as found by Smith (1964)', and if not, it should read like '. . . was formerly shown (Smith, 1964)'. Might be incorrect use of LaTeX commands \cite{} and \citep{}.

References with two authors are cited like (Ulaby and Long 2014), not like (Ulaby et al. 2014).

Abstract and main text should be in present tense, not past tense.

Overall, I suggest accepting the paper after major revisions.

Other points:

Page 2 L(ines) 12-15:roughness explanation too short to be understandable without further reading. Some questions: 'high pass filtering (cut off at 0.25m)': high pass filtering occurs in frequency domain, but you give a length as cut off.

Give Fraunhofer criterion explicitly to make manuscript understandable without further reference.

P5L11 the current version of MILLAS takes into account multiple reflections: if this is new, then describe it in more detail.

Fig. 2: indicate which columns are used for the three curves in Fig 3, e.g. by using the same colors as in Fig. 3.

Fig. 3: give average values of slope, and give slope in deg instead of rad.

Fig 4: indicate the values used for the three curves in Fig 3, e.g. by corresponding colors.

P9L1: which is the direction of Phi_0: North? Flight direction?

P9L5: "local" coordinate system is an unhappy name, as all coordinate systems introduced are centered at the footprint center. Suggestion: we introduce a tilted coordinate system with the same origin, but the z-coordinate aligned .. with ^n_i.

Eq. (9): define A, R.

Fig. 5: T_B H/V reads like a ratio, better call it e.g. T_B H,V. Explain ITS, CDF_alpha

[Figure]

If formula symbols are use in text, omit the article: Instead of '..the theta is the incidence angle and the phi is the azimuth..' say '..theta is the incidence angle and phi is the azimuth.. Occurs many time through whole text.

Minor points:

P(age) 1, L(ine) 11: take out incorrect blanks: 'on surface permittivity, second . . .'

P2L9: The incident wavelength reacts differently with individual components of the superimposed roughness: 1. Do you mean The incident radiation ? 2. Term superimposed roughness unclear. Do you mean roughness at different scales?

P3L30: 30% RFI contamination: in time or in signal energy?

P4L9: vertical, horizontal or both?

P4L16: define ALS

P5L23 boned -> beyond

P5L33 Reference: do not give first names, check bibtex file

P8L9 "global" coordinate system in Cartesian basis (..) → Cartesian coordinate system with the origin in the center of the sensor footprint

P12 end of L9: end → and

P13L3: height → high

P13L23: Figures 7,8, → Figures 7 and 8

P15L3: We want to determine the simulation setup that best reproduces . . ..

P16L8: I do not find 4.5 K in Table 1. Do you mean 4.6 K?

P17L9: decrees → decreases

P17L10: decreased → decreases, increased → increases

P17L13: ..strongest for the roughest surface

P18L5: had → has

P18L7: inclusion of a crude snow. . .; A possible explanation. . .

P18L11: the microphysical snow and sea ice properties

P18L13: on request

---

## Author Comment (AC2) · 15 Sep 2019

Dear Dr. Heygster,

I am very grateful for you time that you took to analyze the manuscript and provide the remarks. Following the reviewers suggestions I included additional subsections in the manuscript. Section 2 now contains the subsections dedicated to the analysis of the facet orientation angles: the slope and the facet azimuth direction. Section 3 is now supplemented with sensitivity analysis of the model. To put the results in the broader context we reflect on the implication on the SMOS sea ice thickness product, as well as on the planned CIMR mission. According to your suggestions we refrazed some of

the sections to improve readability of the manuscript.

We took extra care to address all your remarks, nonetheless if some points require further attention we will gladly provide more exhaustive response.

Regarding your remarks to the manuscript.
   **The discussion of Fig. 6 and its use for interpretation of the experiments is**

**incomplete:One of the interesting results of Fig. 6 is that between 4045angle, the h-pol TBs are practically insensitive to the roughness parameter s. This is important for L band satellite sensors observing only at such incidence angles like SMAP (in orbit since 2015) and the upcoming CIMR, and for the airborne observations at 45°(Figs. 9 and 10, Table 1): In the case when no influence of the roughness on the TB signal is expected (h-pol), the found correlation between observation and model is clearly higher, the RMSE, bias and the ubRMSE all are higher than at v-pol, where the model predicts a sensitivity to roughness. In Figs. 9(c) and 10(c), the h-pol 45angle cases, the modeled TBs show clearly less variability than the corresponding v-pol cases (Figs. 9(d) and 10(d)). Do you have an interpretation for this finding?**

»>The comparison between the simulated and measured TB especially for the slide-looking antenna is not conclusive. First of all, we do not have the roughness information from the slide-looking footprint. We assume that the statistical distribution of facet geometries are the same as for the nadir one. Secondly, the assumption of the constant antenna gain over field of view is not adequate. The Geometrical roughness model describes the sub-footprint characteristics of the sea ice surface. Unfortunately, we do not have the measurements of the antenna gain functions when mounted on the aircraft. Thirdly, as indicated by the sensitivity the simulated TB is much more sensitive to the snow cover than to the surface roughness. However, due to the lack of independent snow measurements we resort to making assumptions about its thickness.

[Figure]

**Fig. 6 and P12 L9-10 '..the horizontal and vertical polarization curves are brought together.' Correct only at incidence angles > 45°. At lower angles, the opposite is the case. Best, add to Fig. 6 the polarization difference curves near the bottom, potentially at an increased y scale.**

»>The polarization difference is now added as a subplot the Figure 6.

**Fig.6: Give x axis in degree, not in rd because in text you use deg.** »>All angles are now given in degrees.

**The current version of eq. (3) contains a product instead of a sum ('+' missing), and eq. (5) is incorrectly copied from Ulaby and Long, (2014), p443: replace $\hat{r}$ in nominator and in denominator by $\hat{n}$, and check order of factors. $\hat{r}$ in this equation does not make sense at all:$\hat{y}$ should be independent of $\hat{r}$ !**

»>Thank you for double checking the equation 3 (eq. 4 in the current version of the manuscript). It is a typo, the antenna looking direction $\hat{r}$ is looking downwards, therefore a "-" in the z axis is added.

Equation 5 after Ulaby and Long 2014 p443 , in the book the $n_i$ denotes the "direction of propagation" , which in our case is the antenna look direction" $\hat{r}$. Probably my change of the notation contributed to confusion. In the manuscript the subscript "i" is reserved for the facet coordinates: $n_i$ - facet normal, $y_i$ and $x_i$ coordinates on the facet's plane.

**Units should be given in a consistent way throughout the whole manuscript. Here, the units m, cm and mm all occur, which is confusing and makes reading cumbersome. Have always a blank between number and units.** »>Units: all the distance units are now converted to meters and all units are formatted with the siunitx package, (eg. 273K)

**The references in the text are frequently odd: if part of a sentence, then it should read 'as found by Smith (1964)', and if not, it should read like '...was formerly shown (Smith,1964)'. Might be incorrect use of LaTeX commands cite and citep. References with two authors are cited like (Ulaby and Long 2014), not like (Ulaby et al.2014).** »>Citation formatting was corrected with more consistent use of "citep" and "citet" commands.

**Abstract and main text should be in present tense, not past tense.Overall, I suggest accepting the paper after major revisions** »> The paragraphs are re-written in present tense.

**Other points  Page 2 L(ines) 12-15:roughness explanation too short to be under-**

**standable without further reading. Some questions: 'high pass filtering (cut off at 0.25m)': high pass filtering occurs in frequency domain, but you give a length as cut off.**

»> With reference to the spatial frequency in L7 i changed the frequency unit to mâĄż1

**Give Fraunhofer criterion explicitly to make manuscript understandable without further reference.** »>The Fraunhofer criterion is now explicitly stated

**P5L11 the current version of MILLAS takes into account multiple reflections: if this is new, then describe it in more detail.** »>An explanation about multiple reflections in MILLAS is added.

**Fig. 2: indicate which columns are used for the three curves in Fig 3, e.g. by using the same colors as in Fig. 3. Fig. 3: give average values of slope, and give slope in deg instead of rad.Fig 4: indicate the values used for the three curves in**

**Fig 3, e.g. by corresponding colors.** »>The color coding is introduced on figure 2, and subsequent. **P9L1: which is the direction of Phi_0: North? Flight direction?**

»> the coordinates system is defined such as $\Phi_0 = 0$, ie. so as $\hat{x}$ is parallel to the ground and on the plane including the nadir and side looking antenna directions.

**P9L5: "local" coordinate system is an unhappy name, as all coordinate systems introduced are centered at the footprint center. Suggestion: we introduce a tilted coordinate system with the same origin, but the z-coordinate aligned .. with $\hat{n}_i$.**

»> As suggested, we reformulated. Now we use 'tilted' instead of "local".

**Eq. (9): define A, R.** »>Eq9 Explanation added, A - facet area, R - distance 'antenna-facet'

**Fig. 5: T_B H/V reads like a ratio, better call it e.g. T_B H,V. Explain ITS, CDF_alpha** »>Fig 5 Notation changed $T_B H, V$ instead of $H/V$, explanation of ITS and CDF is now added to the title.

**Regarding "Minor points" P(age) 1, L(ine) 11: take out incorrect blanks: 'on**

**surface permittivity, second...'**

»>P1L11 blanks taken out

**P2L9: The incident wavelength reacts differently with individual components of the superimposed roughness: 1. Do you mean The incident radiation ? 2. Term superimposed roughness unclear. Do you mean roughness at different scales?**

»> Rephrased, now: . . ."The incident radiation of a given wavelength reacts differently

with individual components of the superimposed roughness of many scales...”

**P3L30: 30% RFI contamination: in time or in signal energy?** »> 30% RFI contamination refers to the number of samples, this is now added to text.

**P4L9: vertical, horizontal or both?** »> This is unclear to me, as it refers to the Airborne Laser scanner description.

**P4L16: define ALS** »> the ALS abbreviation is now explained

**P5L23 boned -> beyond,** »>Now corrected

**P5L33 Reference: do not give first names, check bibtex file** »>Checked, and corrected

**P8L9 "global" coordinate system in Cartesian basis (..)Cartesian coordinate system with the origin in the center of the sensor footprint**

»>Changes as suggested

**P12 end of L9: end→and**

»> corrected

**P13L3: height→high**

»> corrected

**P13L23: Figures 7,8,→Figures 7 and 8**

»> corrected

**P15L3: We want to determine the simulation setup that best reproduces....**

»> corrected

**P16L8: I do not find 4.5 K in Table 1. Do you mean 4.6 K?** »> Yes, corrected

**P17L9: decrees→decreases**

»> corrected

**P17L10: decreased→decreases, increased→increases**

»> corrected

**P17L13: ..strongest for the roughest surface**

»> corrected

**P18L5: had→has**

»> corrected

**P18L7: inclusion of a crude snow...; A possible explanation**...

»> corrected

**P18L11: the microphysical snow and sea ice properties**

»> corrected

**P18L13: on request**

»> corrected

Please also note the supplement to this comment:
https://www.the-cryosphere-discuss.net/tc-2019-110/tc-2019-110-AC2-
supplement.pdf

[Figure]

**Supplement:**

**Effects of decimetre-scale surface roughness on L-band Brightness Temperature of Sea Ice**

Maciej Miernecki[2,1], Lars Kaleschke[3,1], Nina Maaß[1], Stefan Hendricks[3], and Sten Schmidl Søbjærg[4]

[1]Institute of Oceanography (IfM), University of Hamburg, Bundesstr. 53, 20146 Hamburg Germany

[2]Centre d'Etudes Spatiales de la Biosphère (CESBIO), 18 avenue Edouard Belin bpi 2801, 31401 Toulouse Cedex 9, France

[3]Alfred Wegener Institute, Helmholtz Centre for Polar and Marine Research, Bremerhaven, Bussestrasse 24, 27570 Bremerhaven, Germany

[4]Technical University of Denmark, Ørsteds Plads, 2800 Kgs. Lyngby Danmark

**Correspondence:** Maciej Miernecki (maciej.miernecki@cesbio.cnes.fr)

**Abstract.** Sea ice thickness measurements with L-band radiometry  allow for daily, weather-independent monitoring of the polar sea ice cover. The sea-ice thickness retrieval algorithms  rely on the sensitivity of the L-band brightness temperature to sea-ice thickness. In this work, we investigate the decimetre-scale surface roughness as a factor influencing the L-band emissions from sea ice. We  use an airborne laser scanner to construct a digital elevation model of the sea ice surface. We  find that the probability density function of surface slopes is exponential for a range of degrees of roughness. Then we  apply the geometrical optics,  bound with the MIcrowave L-band LAyered Sea ice emission model in the Monte Carlo simulation to simulate the effects of surface roughness. According to  these simulations, the  vertical polarization around Brewster's angle  is most affected by decimetre-scale surface roughness with brightness temperature decreasing up to $8\,\mathrm{K}$. The horizontal polarization for the same configuration exhibits a $4\,\mathrm{K}$ increase. The near-nadir angles are little affected, up to $2.6\,\mathrm{K}$ decrease for the most deformed ice. These result indicate that the current operational sea ice thickness retrieval algorithm using the near-nadir L-band is marginally affected by omission of the surface roughness. Overall the effects of large-scale surface roughness can be expressed as a superposition of two factors: the change in intensity and the polarization mixing. The first factor depends on surface permittivity, the second shows little dependence

on it. The sensitivity analysis indicates that snow cover impacts the brightness temperature to a greater extent than surface roughness. Comparison of the brightness temperature simulations with the radiometer data does not yield definite results.

*Copyright statement.* TEXT

**1 Introduction**

[revised manuscript text omitted]
 surface permitivities. ice temperatures ($250\,\text{K}$ $< T_{ice} < 271\,\text{K}$) and salinities ($1\,\text{g/kg} < S_{ice} < 12\,\text{g/kg}$).

The sea ice permitivities from the MILLAS model range between $\epsilon = 3.1 + 0.05i$ (for $T_{ice} = 271\,\text{K}$, $S_{ice} = 7\,\text{g/kg}$) and $\epsilon = 4.6 + 0.8i$ (for $T_{ice} = 253\,\text{K}$, $S_{ice} = 1\,\text{g/kg}$), where $T_{ice}$ is the bulk ice temperature and $S_{ice}$ the bulk ice salinity. The curves corresponding to those values lie close together indicating that the proposed parametrization is suitable  for all types of  sea ice.  The effect of permittivity on the polarization mixing parameter ($Q_\alpha$) is less pronounced. The dependence of the $Q_\alpha$ parameter on the roughness follows a similar quadratic curve regardless of the surface permittivity.

**3.2 Sensitivity analysis**

In this section, we investigate the sensitivity of our model. This step will enable the interpretation of the results of the comparison between simulations and measurements presented in section 3.3. We start by estimating the sensitivity to sea ice contrition. Than we progress to analyze the model inputs: surface temperature, ice thickness, snow thickness, surface roughness.

The two most important factors influencing the L-band brightness temperature over sea ice are the ice concentration and the ice thickness. We calculate the sensitivity of our model to sea ice concentration by assuming a linear mixing of water and thick ice fractions within the radiometer footprint. The brightness temperature of sea water $T_{Bw}$ (salinity of $33\,\text{g/kg}$, temperature $271.2\,\text{K}$) is approximately $110\,\text{K}$ and $T_{Bi}(0, p/q)$ of thick sea ice ($T_{surf} = 260\,\text{K}$, $d_{ice} = 1.5\,\text{m}$, bulk salinity of $3\,\text{g/kg}$) is $240\,\text{K}$. The resulting sensitivity to sea ice concentration is $\approx 1.5\,\text{K/\%}$.

The sensitivity of the $T_B$ to sea ice thickness over thin sea ice $d_{ice} < 0.75\,\text{m}$ is fundamental for the sea ice thickness retrieval from L-band radiometry. It is only when the sea ice thickness is significantly larger than the penetration depth of the L-band radiation when the $T_{Bi}$ saturates. Therefore, in order to single out the contributions of surface roughness, our analysis is concentrated on sea ice thicker than $1\,\text{m}$.

Table 2 contains the sensitivities of the geometrical roughness model to the input parameters: roughness parameter $s_{alpha}$, ice thickness $d_{ice}$, snow thickness $d_{snow}$, surface temperature $T_{surf}$. Presented values are grouped into columns corresponding to the polarization and three incidence angles: $0°$, $45°$ and $60°$. The angles where chosen to reflect the antennae configuration during the SMOSice2014 with an additional $60°$ close to Brewster's angle where surface roughness effects are most pronounced.

Figures 7, 8 show the simulated L-band brightness temperature at $0°$, $45°$ as a function of the surface temperature. In presented study we make considerable assumption about snow thickness. To illustrate the assumptions the plots are made for a range of snow thicknesses in corresponding line styles. In the MILLAS model, ice permittivity is parameterized with ice temperature. The non-monotonic shape of the curves is caused by change in ice permittivity. Therefore, in table 2 the relevant values of sensitivities are given for lower ($250\,\text{K}$-$265\,\text{K}$) and higher ($265\,\text{K}$-$270\,\text{K}$) temperature ranges.

The assumption about snow thickness has a considerable effect on the sea ice $T_B$ (Maaß et al., 2013). For this reason the values of sensitivities are considered for a number of snow thickness values.

As far as the large-scale surface roughness is concerned the sensitivity increases almost linearly for the values of $s_\alpha$ between 0° and 20° which is the maximal value measured during the SMOSice2014 campaign.

[Figure]

**Figure 7.** Change in the  nadir brightness temperature as predicted by the MILLAS emission model as a function of surface temperature.  The different  line styles correspond to the different snow thickness assumptions. The calculation was done for sea ice thickness of one meter and surface roughness set to zero.

In order to interpret the results of the simulation-measurements comparison it is necessary to evaluate the uncertainties associated with the input parameters: surface temperature, ice thickness and snow thickness. In the following paragraphs we by "mean model sensitivity for the cold conditions" we understand the averaged absolute sensitivity for $T_B(0; H, V)$ and $T_B(45; H, V)$ at $250\,\mathrm{K}$. We take the values for the lower temperature rage as they reflect the conditions during SMOSice2014 campaign.

The surface temperature measurements done with the KT19.85 have an accuracy of $0.5\,\mathrm{K}$. The mean surface temperature in the region covered by ice was $251.7 \pm 3.5\,\mathrm{K}$. We take the standard deviation of surface temperature measurements as the parameter uncertainty. Then we multiplied the parameter uncertainty by average absolute model sensitivity at low temperatures to obtain the model uncertainty associated with it. Thus, we estimate the uncertainties associated with surface temperature is $0.7\,\mathrm{K}$.

[revised manuscript text omitted]

---

## Author Comment (AC3) · 31 Oct 2019

Dear Dr. Landy,

I am writing to you to supplement my initial response to your remarks. In the most recent version of the manuscript (2019/10/31) we used the measured, empirical CDFs instead of the fitted ones. Also, in the attachment you will find the figures with the exponential fits to the slopes PDFs for all of the roughness classes (FIg. 1 and Fig. 2 added below).

[Figure]

Kind regards,

Maciej Miernecki

Remakes considered:

**4. Could you not just use the observed empirical CDF within each 70 m footprint, rather than the statistical model fit, to simulate Tb? i.e. integrate over the N pairs of angles for each facet within the 70 m footprint. Is this just to speed up your simulations($70^2/0.5^2$ is only about a factor 2 larger number of facets than your $10^4$ criteria), or so you can calculate average model results over 5 km sections? Using the observed CDF may produce a better model fit to the radiometer observations.**

»> In the current version of the manuscript we use the empirical CDF directly from the measurements (without fitting). This indeed improved the r2 performance matrix. However, the simple one-layer model setup that we used is inadequate to simulate the sea ice brightness temperature.

**P7 L10. Can you show the exponential function fit to each class of data in Fig 3, so we can see how well it performs?**

»> In the attached files and FIg. 1 and Fig. 2 shown bellow, we present the exponential fits to the slopes PDF for all of the roughness classes. Although in the manuscript (Figure 4) we have added only the fits for the selected ones.

Please also note the supplement to this comment:
https://www.the-cryosphere-discuss.net/tc-2019-110/tc-2019-110-AC3-supplement.pdf

[Figure]

Fig. 1.

[Figure]

Fig. 2.

**Supplement:**

---

## Author Response (AR1)

In this document we make a point-by-point response to the reviewers remarks. We also provide the marked-up version of the submitted manuscript highlighting the changes. According with the reviewers suggestions new sections and analysis are added. The manuscript was also partially re-written.

**Response to Reviewer 1**

General comments:

**1. It is not obvious from the paper what are the implications of your results for sea ice thickness measurements from satellite, e.g. SMOS or the upcoming CIMR mission.What is the relative importance of decimetre roughness compared to other factors? Do the current incidence angles employed by SMOS limit the sensitivity of measured Tb to roughness? I expected to see a statement on this in the abstract and some discussion later on the manuscript.**

≫ The discussion is now included in the manuscript: The SMOS SIT is derived from near-nadir measurements (0-30degrees) therefore the expected change in the TB due to the large scale surface roughness (up to -2.6K) is negligible compared to the uncertainty associated with other factors, such as sea ice concentration (-1.5K/%), and snow cover (8.5K/m). The effects of surface roughness are most pronounced for incidence angles >40deg and stronger on vertical polarization. We have added the statements regarding that current and future L-band missions (SMOS, SMAP, CIMR, SMOS-HR) that can be affected.

**2. The airborne altimeter data provide measurements of the snow surface roughness, but this is not necessarily reflected directly in the underlying ice-snow interface roughness. There is no discussion of this in the manuscript and the potential issues/errors it could introduce. Which is most important for L-band emissivity, snow or ice surface roughness? Might the roughness be overestimated if it's the ice interface roughness that you need to know?**

≫ The altimeter measures the snow surface elevation but we lack snow thickness data. Therefore we have to assume that snow is plane-parallel layer to the sea ice surface. Although dry snow is transparent in L-band it has an impact on ice thermodynamics, which determine the ice effective temperature and emissivity (snow layer also refracts the radiation, but it is more relevant for higher incidence angles).

Regarding the overestimation of the sea ice roughness due to snow cover. We would rather expect the opposite effect, the underestimation of the surface roughness. Snow will cover the ragged edges of the pressure ridges and ice flows.

In the sensitivity study we calculate the model sensitivities for different assumptions about snow thickness. The importance of the snow thickness is now illustrated on two new figures: 8 and 9. The change of TB due to the snow thickness is up to 18K (from 0 to 1m) for sea ice surface temperature of 250K.

**3. It is not clear whether the assumption of isotropically-oriented surface roughness features is valid, even when averaging model-data comparisons over 5 km. Is the sensitivity of modelled Tb to surface feature orientation linear? When modelling Tb over 5km, the assumption is that Tb will be the average of a uniform distribution of surface feature azimuth angles. But is it reasonable to assume the average of short radiometerTb integrations, from sea ice with lots of different surface feature orientations, is mea-suring the same thing? Is there any geometrical shadowing of facets at the 45-degree incidence angle? If so, how do you account for this in the simulations?**

≫ Regarding the isotropic orientation of surface facets. We now include the analysis of the surface slopes azimuthal orientation: figure3. Although the comparison with the radiometer data is not conclusive at 5km, we aim at generalizing the roughness parametrization for a larger region, so as to infer the implications for SMOS/SMAP/CIMR (resolution of 40km), at such scale it is less likely that the sea ice will have coherent ondulations on the surface. The impact of such oriented sinusoidal surfaces on the angular characteristics is discussed in Ulaby and Long (2014) in chapter 10.4., which served as an inspiration for our approach.

In the simulation the shadowing occurs when the local incidence angle is greater than 90deg, therefore radiation from such facet is emitted away from the antenna. Current version of the statistical model, with the facet orientation drawn from the CDF, does not account for the "double-bounce" reflections.

**4. Could you not just use the observed empirical CDF within each 70 m footprint,rather than the statistical model fit, to simulate Tb? i.e. integrate over the N pairs of angles for each facet within the 70 m footprint.**

**Is this just to speed up your simulations($70^2/0.5^2$ is only about a factor 2 larger number of facets than your $10^4$ criteria), or so you can calculate average model results over 5 km sections? Using the observed CDF may produce a better model fit to the radiometer observations.**

»> We considered the direct assimilation of the facet orientation from the DEM but this method will be only applicable to the nadir antenna as the side-looking footprint is not scanned. In the manuscript version submitted on 2019/10/31 we use the empirical CDFs derived directly from the ALS measurements.

Another associated issue is the assumption on the ice thickness and snow thickness of each facet. Although these can be substituted with constant values for the whole footprint. Additionally, the position of the facets within the antenna gain function, which is unknown for this particular system mounted on the airplane. The assumption of the constant gain that we used in the manuscript is more accurate as the signal is averaged over a larger distance. All things considered, the assumptions that have to be made and rather small expected signal (up to 8K) directed our efforts to establish a simple robust parametrization which considers an isotropic slope orientation.

**5. A particularly useful contribution of this paper would be a more in-depth model sensitivity analysis of the relative effect of roughness on measured Tb compared to other factors. The current results touch on this with e.g. Fig 6, but by keeping other factors constant in the simulations its impossible for the reader to understand the true sensitivity to roughness. For example, how different does Fig 6 look for a different set of sea ice constants? E.g. 3 m thick, fresh MYI, with thicker snow depth and a warmer surface? I would recommend removing Figs 7 and 8, which don't really contribute to the message of the paper, and adding some deeper theoretical analysis of the relative impact of roughness on L-band Tb.**

»>Thank you for this suggestion, the sensitivity analysis is now included in the manuscript. Based on it we present that surface roughness has a much smaller impact on the L band TB of sea ice than snow cover. New figures: Fig 8a, b, c show the non-linear and non monotonic relation between TB and surface temperature, plotted for different snow thicknesses. Table 2. Contains the partial sensitivities to of the TB to the various inputs, for lower and higher temperature ranges as well as for set of assumptions on snow thickness. The previous figures 7, 8 are removed.

**6. Results from the comparison between modelled and observed Tb are not promising and are difficult to interpret here. I had many questions looking at Figs 9 and 10 that were not discussed within the text. Based on the theoretical results in Fig 6, you'd probably only expect an improvement to the 45-degree angle v-pol channel when in-cluding the effects of roughness, right? So the poor correspondence between model and observations is likely one or more of: model inaccuracy, the model configuration(no. layers, penetration etc.) not being ade-quate, simple treatment of ice thermody-namics, not having altimeter observations for the 45-degree footprints, or the limited treatment of snow. Without showing results from a model sensitivity analysis of these factors though, it is impossible to interpret which factor or set of factors is most likely.Why is there such a low dynamic range for modelled Tb's in most cases where mea-sured Tb >220-240K? Can you add another figure showing the absolute differences between modelled Tb for simulations with and without GO roughness included, per-haps as histograms or as a function of the surface roughness?**

»> The discussion of the comparison between modeled and measured TB is now interpreted with the sensitivity study in mind. The simple emission model used in our study (one layer of ice with one layer of snow) has much more sensitivity to the snow thickness than to the surface roughness. We added new figures showing the histogram of the differences between measurements and the simulations setups for all four antenna feeds. Overall, the simplified one-layer model poorly simulates the sea ice TB.

**7. The written English needs some improvement throughout the manuscript. I would recommend a careful proof-read to check spelling and grammar. A few e.g.'s just on the first page are:**

**L2 'rely on',** »> corrected

**L8 you mean 'horizontal polarization'?** »> corrected

**Minor comments/edits:**

**Page 1. Line 7. Effect on what? What scale of roughness? Multiple scales?**

»> corrected

**L 18. Surface roughness of the ice or snow, or both?**
»> corrected

**P2 L5. Both high and low spatial frequencies…**
»> corrected

**L11. What do you mean by 'stays unnoticed'? Rephrase**
»> rephrased to: the surface roughness in negligible

**L18. What is 8*lambda for L-band?Intro Section. What about the other factors affecting Tb from sea ice? They are not the primary focus of this study, but can complicate your interpretations of the roughness effects, particularly when comparing model results to radiometer observations. So you should introduce the effects of e.g. thermodynamics, ice concentration, snow proper-ties etc. here. Are there any previous estimates of the impact of roughness on L-bandTbs?**
»> There are a number of studies regarding effects of surface roughness on the L-band TB of soils, as this wavelength is widely used for soil moisture retrieval. To our knowledge this is the first time that the impact of surface roughness is on L band emissions from sea ice is evaluated.

**P3 L8-9. What was the air temperature then? Thermodynamic effects on the snow properties may explain your difficulties comparing model and observations, and the large impact of a snow layer on your simulations then?**
»> During the 24. of March The average Sea ice surface registered by the KT19.85 was 251.7+/-3.5K. As the sensi-tivity analysis indicate, the assumption on snow thickness being a constant fraction of the sea ice thickness is a major source of uncertainty.

**P4 L1-3. Can you provide a little more detail on this as its such a substantial bias?How do you know it was purely additive? How was this tested?**
»>According to the campaign report Hendricks et al.2014 the bias was determined and corrected by taking measure-ments over open water and performing so called wing wags to cross calibrate the two antennae feeds. Also the RFI filtration system based on frequency and time sampling masked out the samples that were showing characteristics of an artificial source.

**P4 L10. Here I found myself asking if you used the same roughness data from nadir to simulate the 45-degree return. This was answered much later but you should state it here.**
»> corrected

**P5 L2-3. How was the sea level estimated from lead tie-points? Do you have uncertainties for the sea level, freeboard and ice thickness, that you can apply to estimate uncertainties in modelled Tb? How did you estimate snow depth uncertainty when applying a simple snow scheme?**
»> According to the campaign report Hendricks et al.2014, the tie points were picked manually giving the reference for the elevation measurements. The sea ice thickness is calculated from re-sampled to 1second ALS data, we took the standard deviation of the 1s sample as uncertainty. We estimate the snow thickness as a constant fraction of the sea ice thickness, thus we assume a propagation of uncertainty derived from elevation uncertainty.

**L4. Do you use an iterative procedure to estimate ice thickness then, if the thickness is already required to estimate snow depth?**
»> Unfortunately not, we assume a hydrostatic equilibrium and that the snow thickness is 1/10 of the ice thickness.

**L11. Do you have a citation for the version of MILLAS used here, or was this added work completed as part of your study? If the latter, you need to describe model addi-tions including equations for review (perhaps in an appendix).**
»> corrected. We included a short description in the manuscript.

**L16. How are the ice and water salinities calculated? (You need to make it clear here that you use the ice sur-face T from airborne observations to constrain ice thermody-namics in the simulations). I'd like to see a table here of the constants used for ice,water and snow physical parameters, and then the range over which other**

**parameters(e.g. roughness) varied.** »>Thank you for this suggestion, we include the formulas and constants used in MILLAS in the Table 1

**L30-31. It's important here that you state ALS observations of roughness are averaged over quite a long window. As it's currently written, it sounds like you are simulating and comparing with measured returns over the exacts same 70 m window.**

**L34. Use a proper citation style for this reference.**
»> corrected

**P6 L4-5. Did you filter out all 70-m sections containing mixed classes, e.g. some open water? What about thin leads within the footprint? I'd expect many of your eventual 5-km sections contained at least some open water, so how was this accounted for?**
»> We compute the surface roughness statistics from 70m section treating it as a one class. We exclude from the analysis the 1s sections with more than 5% missing data. This rather crude method assumes that thin ice or open water will not reflect the laser scanner resulting in missing data. An alternative would be to use the camera images, but those were not taken continuously.

**L13. 'coast'** »> corrected

**P7 L1-2. I'd like to see a figure which proves this. This cutoff limit between anisotropic and isotropic orientation of surface features has not been shown before, so a novel result of this study. But if you want to prove there is a scale separation at 4.3 km you need to show the data**
»> New section: Section 2.2.1 no describes the azimuth orientation of the facets along the flight track with the figure

**L10. Can you show the exponential function fit to each class of data in Fig 3, so we can see how well it performs? Eq 9. What is R?**
»> R is the distance from facet to the antenna, explanation is now added together with the fits. Although, we show three example of smooth; medium and rough ice as placing all the classes cluttered the figure.

In the latest version of the manuscript (submitted on 2019/10/31) we add the exponetial fits to the PDFs for the selected classes. (see figure 4). In the figures bellow we present the fits for all roughness classes.

[Figure]

[Figure]

**P10 L3-4. Is it reasonable to assume a constant gain pattern over the entire FOV? Do you have an estimate of the antenna pattern to compare to?**
»> We do not have the antenna pattern for the setup during the campaign. It is possible to make assumptions about it based on the horn size, however when mounted on the plane the sidelobes and gain will vary. When considering the case of isotropic slope orientation the antenna gain can be considered constant.

**L10. So the T profile is calculated directly from surface T and the reference constant salinity?**
»> Yes, according to the formulas presented in Table 1: (after Untersteiner, 1964) Snow thermal conductivity = 0.31W/(mK) Ice thermal conductivity = 2.034 W/(mK)+0.13W/m * Sice(g/kg)/Tice(K) Ice salinity = 4 g/kg P12 L5. If you refer to angles in degrees within the text, the x-axis in Fig 6 should also be in degrees

**L9. 'And'?** »> corrected

**L12-13. Confusing. What do you mean by this?**

**P13 L3. 'High'?** »> corrected

**P13 L16-31. I can't understand why this section is included, along with Figures 7 and 8.Why not just calculate reasonable variations in MILLAS emissivity for different sea icescenaios? E.g. warm/cold ice, different salinities, shallow/deep snow, different snowT or densities? Relative permittivities up to 10-20 are unrealistic for sea ice in most conditions, so they are not helpful for your analysis here. There's only really reason to show the cases listed directly on Line 24.**

»> corrected, The section is reedited.

**P15 L5. State this earlier in the method.**

**L7. How do you decide when it is needed?**
»> corrected, refrazed

**L8-9. Is the roughness CDF calculated from all altimeter observations within this 5-km window then?**
»> yes. The CDFs are computed from all altimeter observations within 1s section that have less than 5% missing data.

**P18 L8. Unlikely permittivity but possibly thickness. What about open water within footprints? Could that have affected the radiometer measurements? Or maybe snow depth/property variations along track?**
»> The missing data criterion from previous point partially solves the problem of smooth thin ice or open water within the footprint. However, the sensitivity to open water is much greater than that to the surface roughness. Corrected to "...heterogeneous in terms of its thickness"

**L9-10. You at least had the facet orientation info at least for the nadir looking antenna right?** »> yes

**Response to Reviewer 2**

Regarding your remarks to the manuscript.

**The discussion of Fig. 6 and its use for interpretation of the experiments is incomplete:One of the interesting results of Fig. 6 is that between 40and 45deg inc angle, the h-pol TBs are practically insensitive to the roughness parameter s. This is important for L band satellite sensors observing only at such incidence angles like SMAP (in orbit since 2015) and the upcoming CIMR, and for the airborne observations at 45deg(Figs. 9 and 10, Table 1): In the case when no influence of the roughness on the TB signal is expected (h-pol), the found correlation between observation and model is clearly higher, the RMSE, bias and the ubRMSE all are higher than at v-pol, where the model predicts a sensitivity to roughness. In Figs. 9(c) and 10(c), the h-pol 45inc angle cases, the modeled TBs show clearly less variability than the corresponding v-pol cases (Figs. 9(d) and 10(d)). Do you have an interpretation for this finding?**
  »The comparison between the simulated and measured TB especially for the slide-looking antenna is not conclusive. First of all, we do not have the roughness information from the slide-looking footprint. We assume that the statistical distribution of facet geometries are the same as for the nadir one. Secondly, the assumption of the constant antenna gain over field of view is not adequate. The Geometrical roughness model describes the sub-footprint characteristics of the sea ice surface. Unfortunately, we do not have the measurements of the antenna gain functions when mounted on the aircraft. Thirdly, as indicated by the sensitivity the simulated TB is much more sensitive to the snow cover than to the surface roughness. However, due to the lack of independent snow measurements we resort to making assumptions about its thickness.

**Fig. 6 and P12 L9-10 '..the horizontal and vertical polarization curves are brought together.' Correct only at incidence angles > 45. At lower angles, the opposite is the case. Best, add to Fig. 6 the polarization difference curves near the bottom, potentially at an increased y scale.**
  »The polarization difference is now added as a subplot the Figure 7. (new figure numbering)

**Fig.6: Give x axis in degree, not in rd because in text you use deg.**
  »All angles are now given in degrees.

**The current version of eq. (3) contains a product instead of a sum ('+' missing), and eq. (5) is incorrectly copied from Ulaby and Long, (2014), p443: replace $\hat{r}$ in nominator and in denominator by n, and check order of factors. r in this equation does not make sense at all: y should be independent of $\hat{r}$ !**
  »Thank you for double checking the equation 3 (eq. 4 in the current version of the manuscript). It is a typo, the antenna looking direction $\hat{r}$ is looking downwards, therefore a "-" in the z axis is added.

  Equation 5 after Ulaby and Long 2014 p443 , in the book the $n_i$ denotes the "direction of propagation" , which in our case is the "antenna look direction" $\hat{r}$. Probably my change of the notation contributed to confusion. In the manuscript the subscript "i" is reserved for the facet coordinates: $n_i$ - facet normal, $y_i$ and $x_i$ coordinates on the facet's plane.

**Units should be given in a consistent way throughout the whole manuscript. Here, the units m, cm and mm all occur, which is confusing and makes reading cumbersome. Have always a blank between number and units.**
»Units: all the distance units are now converted to meters and all units are formatted with the siunitx package, (eg. 273K)

**The references in the text are frequently odd: if part of a sentence, then it should read'as found by Smith (1964)', and if not, it should read like '...was formerly shown (Smith,1964)'. Might be incorrect use of LaTeX commands cite and citep. References with two authors are cited like (Ulaby and Long 2014), not like (Ulaby et al.2014).**
»Citation formatting was corrected with more consistent use of "citep" and "citet" commands.

**Abstract and main text should be in present tense, not past tense.Overall, I suggest accepting the paper after major revisions**
»> The paragraphs are re-written in present tense.

**Other points:**

**Page 2 L(ines) 12-15:roughness explanation too short to be understandable without further reading. Some questions: 'high pass filtering (cut off at 0.25m)': high pass filtering occurs in frequency domain, but you give a length as cut off.**
»> With reference to the spatial frequency in L7 i changed the frequency unit to $m^1$

**Give Fraunhofer criterion explicitly to make manuscript understandable without further reference.**
»>The Fraunhofer criterion is now explicitly stated

**P5L11 the current version of MILLAS takes into account multiple reflections: if this is new, then describe it in more detail.**
»>An explanation about multiple reflections in MILLAS is added.

**Fig. 2: indicate which columns are used for the three curves in Fig 3, e.g. by using the same colors as in Fig. 3. Fig. 3: give average values of slope, and give slope in deg instead of rad.Fig 4: indicate the values used for the three curves in Fig 3, e.g. by corresponding colors.**
»>The color coding is introduced on figure 2, and subsequent.

**P9L1: which is the direction of Phi_0: North? Flight direction?**
»> the coordinates system is defined such as $\Phi_0 = 0$, ie. so as $\hat{x}$ is parallel to the ground and on the plane including the nadir and side looking antenna directions.

**P9L5: "local" coordinate system is an unhappy name, as all coordinate systems introduced are centered at the footprint center. Suggestion: we introduce a tilted coordinate system with the same origin, but the z-coordinate aligned .. with $\hat{n}_i$.**
»> As suggested, we reformulated. Now we use 'tilted' instead of "local".

**Eq. (9): define A, R.**
»>Eq9 Explanation added, A - facet area, R - distance 'antenna-facet'

**Fig. 5: T_B H/V reads like a ratio, better call it e.g. T_B H,V. Explain ITS, CDF_alpha**
»>Fig 5 Notation changed $T_B H, V$ instead of $H/V$, explanation of ITS and CDF is now added to the title.

**Regarding "Minor points":**

**P(age) 1, L(ine) 11: take out incorrect blanks: 'on surface permittivity, second...'**
»>P1L11 blanks taken out

**P2L9: The incident wavelength reacts differently with individual components of the superimposed roughness: 1. Do you mean The incident radiation ? 2. Term superimposed roughness unclear. Do you mean roughness at different scales?**
»> Rephrased, now: . . . "The incident radiation of a given wavelength reacts differently with individual components of the superimposed roughness of many scales..."

**P3L30: 30% RFI contamination: in time or in signal energy?**
»> 30% RFI contamination refers to the number of samples, this is now added to text.

**P4L9: vertical, horizontal or both?**
»> This is unclear to me, as it refers to the Airborne Laser scanner description.

**P4L16: define ALS**

»> the ALS abbreviation is now explained

**P5L23 boned -> beyond,**
»>Now corrected

**P5L33 Reference: do not give first names, check bibtex file**
»>Checked, and corrected

**P8L9 "global" coordinate system in Cartesian basis (..)Cartesian coordinate system with the origin in the center of the sensor footprint**
»>Changes as suggested

**P12 end of L9: endand**
»> corrected

**P13L3: heighthigh**
»> corrected

**P13L23: Figures 7,8,Figures 7 and 8**
»> corrected

**P15L3: We want to determine the simulation setup that best reproduces....**
»> corrected

**P16L8: I do not find 4.5 K in Table 1. Do you mean 4.6 K?**
»> Yes, corrected

**P17L9: decreesdecreases**
»> corrected

**P17L10: decreaseddecreases, increasedincreases**
»> corrected

**P17L13: ..strongest for the roughest surface**
»> corrected

**P18L5: hadhas**
»> corrected

**P18L7: inclusion of a crude snow...; A possible explanation…**
»> corrected

**P18L11: the microphysical snow and sea ice properties**
»> corrected

**P18L13: on request**
»> corrected

**Marked-up manuscript**

**Effects of decimetre-scale surface roughness on L-band Brightness Temperature of Sea Ice**

Maciej Miernecki[2,1], Lars Kaleschke[3,1], Nina Maaß[1], Stefan Hendricks[3], and Sten Schmidl Søbjærg[4]

[1]Institute of Oceanography (IfM), University of Hamburg, Bundesstr. 53, 20146 Hamburg Germany

[2]Centre d'Etudes Spatiales de la Biosphère (CESBIO), 18 avenue Edouard Belin bpi 2801, 31401 Toulouse Cedex 9, France

[3]Alfred Wegener Institute, Helmholtz Centre for Polar and Marine Research, Bremerhaven, Bussestrasse 24, 27570 Bremerhaven, Germany

[4]Technical University of Denmark, Ørsteds Plads, 2800 Kgs. Lyngby Danmark

**Correspondence:** Maciej Miernecki (maciej.miernecki@cesbio.cnes.fr)

**Abstract.**

Sea ice thickness  is an Essential Climate
5   Variable. Current L-Band sea ice thickness retrieval methods do not account for sea ice surface roughness that is hypothesized to be not relevant to the process. This study attempts to validate this hypothesis that has not been tested yet. To test this hypothesis, we created a physical model of sea ice roughness based on geometrical optics and merged it into the L-band  emissivity model of sea ice that is similar to the one used in the operational sea ice thickness retrieval algorithm. The facet description of sea ice
10   surface used in geometrical optics is derived from 2-D surface elevation measurements. Subsequently the new model was tested with $T_B$ measurements performed during the SMOSice2014 field campaign. Our simulation results corroborate the hypothesis that sea ice surface  roughness has marginal impact on near-nadir $T_B$ (used in the current operational retrieval). We demonstrate that the probability  distribution function of surface slopes
15

~~around Brewster's angle is most affected by decimetre-scale surface roughness with brightness temperaturedecreasing up to 8 K. The horizontal polarization for the same configuration exhibits a 4 K increase. The near-nadir angles are little affected, up to 2.6 K decrease for the most deformed ice. These result indicate that the current operational sea ice thickness retrieval algorithm using the near-nadir~~ can be approximated with a parametric function whose single parameter can be used to characterize the degree of roughness. Facet azimuth orientation is isotropic at scales greater than 4.3 km. The simulation results indicate that surface roughness is a minor factor in modeling the sea ice brightness temperature. The change in $T_B$ is most pronounced at incidence angles greater than 40 degrees, and can reach up to 8 K for vertical polarization at 60 degrees. Therefore current and future L-band ~~is marginally affected by omission of the surface roughness. Overall the effects of large-scale surface roughness can be expressed as a superposition of two factors: the change in intensity and the polarization mixing. The first factor depends on surface permittivity, the second shows little dependence on it. The sensitivity analysis indicates that snow cover impacts the brightness temperature to a greater extent than surface roughness~~
[revised manuscript text omitted]